# State Space Model Meets Transformer: A New Paradigm for 3D Object Detection

**Chuxin Wang**[1,2]**, Wenfei Yang**[1,2]**, Xiang Liu**[4]**, Tianzhu Zhang**[1,2,3*]
[1]University of Science and Technology of China
[2]National Key Laboratory of Deep Space Exploration, Deep Space Exploration Laboratory
[3]Hainan Aerospace Technology Innovation Center  [4]Dongguan University of Technology
{wcx0602, yangwf}@mail.ustc.edu.cn, liuxiang@dgut.edu.cn, tzzhang@ustc.edu.cn

## Abstract

DETR-based methods, which use multi-layer transformer decoders to refine object queries iteratively, have shown promising performance in 3D indoor object detection. However, the scene point features in the transformer decoder remain fixed, leading to minimal contributions from later decoder layers, thereby limiting performance improvement. Recently, State Space Models (SSM) have shown efficient context modeling ability with linear complexity through iterative interactions between system states and inputs. Inspired by SSMs, we propose a new 3D object DEtection paradigm with an interactive STate space model (DEST). In the interactive SSM, we design a novel state-dependent SSM parameterization method that enables system states to effectively serve as queries in 3D indoor detection tasks. In addition, we introduce four key designs tailored to the characteristics of point cloud and SSM: The serialization and bidirectional scanning strategies enable bidirectional feature interaction among scene points within the SSM. The inter-state attention mechanism models the relationships between state points, while the gated feed-forward network enhances inter-channel correlations. To the best of our knowledge, this is the first method to model queries as system states and scene points as system inputs, which can simultaneously update scene point features and query features with linear complexity. Extensive experiments on two challenging datasets demonstrate the effectiveness of our DEST-based method. Our method improves the GroupFree baseline in terms of $AP_{50}$ on ScanNet V2 (+5.3) and SUN RGB-D (+3.2) datasets. Based on the VDETR baseline, Our method sets a new SOTA on the ScanNetV2 and SUN RGB-D datasets.

## 1 Introduction

With the widespread application of LiDAR and depth cameras, it is becoming easier to obtain 3D point clouds of real scenes. The large amounts of 3D scene data provide rich geometric information for 3D scene understanding in fields such as autonomous driving, robotics, and augmented reality. As a fundamental task in 3D scene understanding, 3D indoor object detection has garnered significant attention from both academia and industry. Unlike 3D object detection (Shi et al., 2019; Yin et al., 2021; Lang et al., 2019; Shi et al., 2020b;a) in autonomous driving scenarios, 3D indoor object detection involves objects with more diverse categories and shapes, posing more significant challenges for the model design and training.

To address the above challenges, numerous 3D indoor object detection methods have been proposed, which can be roughly divided into three categories: vote-based methods (Qi et al., 2019; Xie et al., 2020; Zhang et al., 2020), expansion-based methods (Gwak et al., 2020; Rukhovich et al., 2022; Wang et al., 2022a), and DETR-based methods (Misra et al., 2021; Liu et al., 2021; Wang et al., 2023; Shen et al., 2024). Vote-based methods (Qi et al., 2019; Xie et al., 2020; Zhang et al., 2020) use a voting mechanism to shift surface points toward the object center and then generate candidate points by clustering the points that have shifted to the same regions. Although these methods

---

*Corresponding author.

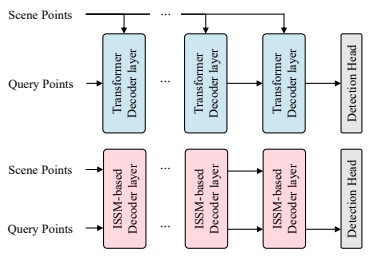 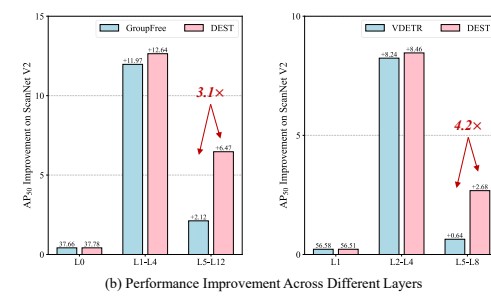

(a) Comparison of Transformer and ISSM-based Decoder Layers  (b) Performance Improvement Across Different Layers

Figure 1: **(a):** Transformer decoder solely updates the features of the query points, while our ISSM-based decoder simultaneously updates the features of scene points and query points. **(b):** The DETR-based models show only slight accuracy enhancements in the later layers, whereas the DEST-based methods significantly boost the performance in the later layers.

have achieved great success in 3D object detection, The vote mechanism is performed in a category-independent manner, leading to the shifted points that are adjacent but belong to different categories being grouped, which limits the model detection capability. Expansion-based methods (Gwak et al., 2020; Rukhovich et al., 2022; Wang et al., 2022a) use generative sparse decoders to generate high-quality proposals based on object surface voxel features with the same semantic prediction. Compared with vote-based methods, expansion-based methods consider the semantic consistency of voxels within the same group and achieve better performance. However, expansion-based methods require a carefully designed proposal generation module and involve numerous manually set thresholds, which limits the model versatility. Recently, DETR-based methods (Misra et al., 2021; Liu et al., 2021; Wang et al., 2023; Shen et al., 2024) have shown promising performance in 3D object detection. Unlike the above methods, DETR-based methods select a small group of voxels or points as the initial object queries and use the scene point features to refine these queries. The query refinement module is simple in design and retains the original geometric structure of the input 3D point cloud. Based on the query refinement module, DETR-based methods have achieved the best performance in indoor object detection tasks. However, DETR-based methods still face a key issue that limits their performance. These methods employ multi-layer transformer decoders (Carion et al., 2020) to iteratively refine the object queries. While the transformer decoder layers update the query point features, they do not simultaneously update the scene point features as shown in Figure 1 (a). As a result, each decoder layer uses the same scene point features to refine the queries, leading to only marginal improvements from the later layers. As shown in Figure 1 (b), we evaluate the detection accuracy improvements of different decoder layers in two DETR-based models (Liu et al., 2021; Shen et al., 2024) on the ScanNet V2 (Dai et al., 2017) dataset. GroupFree (Liu et al., 2021) achieves only a 2.12 performance improvement in the last six layers, while VDETR (Shen et al., 2024) shows a mere 0.64 improvement in the last four layers. Based on the above analysis, the fixed scene point features constrain the potential performance enhancement of the models.

To address this issue, an intuitive idea is to introduce a self-attention mechanism (Vaswani, 2017) between different decoder layers to update the scene point features. However, the quadratic complexity of the self-attention mechanism significantly reduces the model efficiency, making this approach impractical. Recently, state space model-besed methods (Gu et al., 2021a; Gu & Dao, 2023; Dao & Gu, 2024) have shown efficient context modeling with linear complexity through interactions between system states and inputs. The pioneer works lead us to think: **Is it possible to design a State Space Model (SSM) to replace the transformer decoder, enabling the simultaneous update of scene features and query point features?** SSM is used to describe the evolution of system states and to predict future states and system outputs based on system inputs. Therefore, if we model the query point features as the system states and the scene point features as the system inputs at different time steps, we can simultaneously obtain the final system states (updated query point features) and the system outputs at each time step (updated scene point features). However, existing SSMs (Gu et al., 2021a; Gu & Dao, 2023; Dao & Gu, 2024) are not suitable for modeling queries as system states, for two main reasons: **(1). Update the states solely based on the system inputs.** Existing SSMs (Gu & Dao, 2023; Dao & Gu, 2024) adjust the SSM parameters ($\Delta, \mathbf{B}, \mathbf{C}$) solely using system inputs without considering the system states. Therefore, the state points cannot adaptively select system inputs to update themselves, while different query points need to focus on distinct regions of the scene. **(2). Cannot directly process 3D point cloud.** Existing SSMs are inherently designed to process sequential data, and their unidirectional modeling and sensitivity to input order pose challenges for feature modeling of scene points and query points.

Based on the above discussion, we propose a new 3D object DEtection paradigm with a STate space model (DEST) to address the performance limitation caused by the fixed scene point features. The proposed DEST consists of two core components: a novel Interactive State Space Model (ISSM) and an ISSM-based decoder. **In the ISSM**, we model the query point features as the system states and the scene point features as the system inputs at different time steps. Unlike previous SSMs (Gu et al., 2021a; Gu & Dao, 2023; Dao & Gu, 2024), the proposed ISSM determines how to update the system states based on both the system states and system inputs. Specifically, we modify the SSM parameters $(\Delta, \mathbf{B}, \mathbf{C})$ to be dependent on the system states and design a spatial correlation module to model the relationship between state points and scene points. Therefore, the system states in the ISSM can effectively fulfill the role of queries in complex 3D indoor detection tasks. **In the ISSM-based decoder**, four modules are designed for feature modeling of scene points and query points: Hilbert-based point cloud serialization strategy, ISSM-based Bidirectional Scan (IBS) module, Inter-state attention module, and Gated Feed-Forward Network (GFFN). The proposed serialization strategy is designed to serialize the scene points based on the Hilbert curve (Hilbert & Hilbert, 1935), benefiting from its locality-preserving properties. The IBS module is designed to achieve bidirectional interaction among different scene points, while the inter-state attention module is designed to capture the relationships between state points. Lastly, the GFFN is designed to enhance inter-channel correlations through a gated linear unit. The ISSM-based decoder can replace the transformer decoder in DETR-based methods to address the performance limitations caused by fixed scene point features. As shown in Figure 1 (b), our DEST-based method significantly enhances the performance in the later layers.

In summary, the core contributions of this paper are as follows: (1). We propose a novel SSM-based 3D object detection paradigm DEST to overcome the performance limitations caused by fixed scene point features during the query refinement process. To the best of our knowledge, this is the first method to model queries as system states within an SSM framework. (2). We design a novel ISSM whose system states can effectively function as queries in complex 3D indoor detection tasks. In addition, we develop an ISSM-based decoder tailored to the characteristics of 3D point clouds, fully harnessing the potential of the ISSM for 3D object detection. (3). Extensive experimental results demonstrate that the proposed SSM-based 3D object detection method consistently enhances the performance of baseline detectors on two challenging indoor datasets, *i.e.*, ScanNet V2 (Dai et al., 2017) and SUN RGB-D (Song et al., 2015). Moreover, comprehensive ablation studies validate the effectiveness of each designed component.

## 2 RELATED WORK

### 2.1 3D OBJECT DETECTION.

The goal of 3D object detection is to estimate oriented 3D object bounding boxes with their category labels from a point cloud. According to the application scenario, 3D object detection is typically divided into outdoor and indoor detection tasks. Outdoor 3D object detection is commonly used in autonomous driving scenes, where objects are primarily distributed across a wide 2D plane. Therefore, outdoor 3D detection methods typically project the 3D point cloud into a bird's-eye view (BEV) and utilize 2D convolutional networks to detect 3D objects. For instance, MV3D (Chen et al., 2017) directly projects the point cloud onto a 2D grid for feature processing and detection. VoxelNet (Zhou & Tuzel, 2018) first converts the point cloud into a 3D volumetric grid and uses a 3D CNN for feature extraction. Then, it projects the 3D voxels into a BEV for bounding box prediction. PointPillars (Lang et al., 2019) employs PointNet (Qi et al., 2017a) to learn point cloud representations organized in vertical columns (pillars), then uses a 2D convolutional neural network to process flattened pillar features in the BEV.

In contrast, indoor 3D object detection involves handling a more diverse set of object categories and shapes, as well as more complex spatial relationships between objects. Existing indoor 3D object detection methods can be broadly categorized into three groups: vote-based methods, expansion-based methods, and DETR-based methods. For vote-based methods, VoteNet (Qi et al., 2019), as a pioneering work, designs a novel 3D proposal mechanism based on deep Hough voting. ML-CVNet (Xie et al., 2020) introduces three context modules in the voting and classifying stages of VoteNet to encode contextual information at different levels. BRNet (Cheng et al., 2021) backtraces representative points from the voting centers to better capture the fine local structural features sur-

rounding the potential objects from the raw point clouds. H3DNet (Zhang et al., 2020) predicts a hybrid set of geometric primitives and converts the predicted geometric primitives into object proposals. For expansion-based methods, GSDN (Gwak et al., 2020) proposes a generative sparse tensor decoder to generate virtual center features from surface features while discarding unlikely object centers. FCAF3D (Rukhovich et al., 2022) further introduces a fully convolutional anchor-free indoor 3D object detection method. CAGroup3D (Wang et al., 2022a) generates high-quality 3D proposals by leveraging a class-aware local grouping strategy on object surface voxels with consistent semantic predictions. The above methods require carefully designed proposal generation modules and involve several manually set thresholds. DETR (Carion et al., 2020) is a pioneering work that applies Transformers (Vaswani, 2017) to 2D object detection, eliminating many hand-crafted components such as Non-Maximum Suppression (Neubeck & Van Gool, 2006) or anchor boxes (Girshick, 2015; Ren, 2015; Lin, 2017). Currently, DETR and its variants (Zhu et al., 2020; Dai et al., 2021; Liu et al., 2022) have achieved state-of-the-art results in various 2D object detection tasks. Inspired by these works, numerous DETR-based 3D object detection methods have been explored. 3DETR (Misra et al., 2021) is the first to introduce an end-to-end Transformer model for 3D object detection, achieving promising results. GroupFree (Liu et al., 2021) employs a key point sampling strategy to select candidate points and utilizes the attention mechanism to update query point features. LeadNet (Wang et al., 2023) further improves the transformer decoder by introducing a dynamic object query sampling module and a dynamic Gaussian weight map. Most recently, VDETR (Shen et al., 2024) proposes a novel 3D vertex relative position encoding method, which directs the model to focus on points near the object, achieving state-of-the-art performance. However, these DETR-based methods use fixed scene point features in different decoder layers, which limits the detection capabilities of later layers. Unlike the above methods, we propose an ISSM-based decoder that simultaneously updates both scene point and query point features.

## 2.2 State Space Models (SSMs).

Recently, SSMs (Kalman, 1960; Gu et al., 2021a;b) have become a prominent research focus. S4 (Gu et al., 2021a) demonstrates the capability of capturing long-range dependencies with linear complexity. Mamba (Gu & Dao, 2023) further enhances S4 by introducing a selection mechanism, specifically parameterizing the SSM based on the system input. The selection mechanism allows Mamba to selectively retain information, facilitating the efficient processing of long sequence data. Inspired by Mamba, Vision Mamba (Zhu et al., 2024) introduces an SSM-based visual model. VMamba (Liu et al., 2024) furthermore incorporates a 2D selective scan module, enabling the model to perform selective scanning of two-dimensional images. In the field of point cloud understanding, numerous Mamba-based works have emerged. PointMamba (Liang et al., 2024) proposes a simple yet effective Mamba-based baseline, while PCM (Zhang et al., 2024) develops diverse point cloud serialization methods that significantly improve performance. These methods have achieved promising results by leveraging the efficient context modeling and linear complexity of Mamba. Unlike these methods that use SSMs to design feature encoders, we design an SSM-based decoder to address the performance limitations caused by fixed scene point features in the transformer decoder.

## 3 Method

Below, we first briefly review the existing SSMs (Section 3.1), followed by an overview of the proposed DEST (Section 3.2). Subsequently, we offer detailed explanations of the two core components: the Interactive State Space Model (Section 3.3) and the ISSM-based decoder (Section 3.4). Lastly, we outline the model setups for the two baseline models (Section 3.5).

### 3.1 Preliminaries

SSM is used to describe the evolution of system states $h(t) \in \mathbb{R}^K$ and predict future states $h'(t)$ and system outputs $y(t)$ based on system inputs $x(t)$. The system can be defined as follows:

$$h'(t) = \mathbf{A}h(t) + \mathbf{B}x(t), \ y(t) = \mathbf{C}h'(t), \tag{1}$$

where $\mathbf{A} \in \mathbb{R}^{K \times K}$ represents the state transition matrix that describes how the system states evolve, $\mathbf{B} \in \mathbb{R}^{K \times 1}$ denotes the control matrix that describes the influence of the system inputs on the system states, and $\mathbf{C} \in \mathbb{R}^{1 \times K}$ is the observation matrix characterizing the impact of the system states on

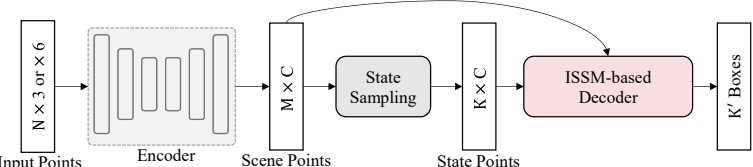

Figure 2: **The overall framework of the DEST-based method for 3D object detection.** We first utilize an encoder to extract 3D features, followed by a state sampling module to select state points, referred to as queries in DETR architecture. Subsequently, we input both the scene points and state points into the ISSM-based decoder for simultaneous updates. Finally, the updated state points are fed into a detection head to predict the 3D bounding boxes.

the system outputs. To handle discrete-time sequence data inputs, the Zero-Order Hold is typically used to discretize the SSM. The discretized version of equation 1 is as follows:

$$h_t = \overline{\mathbf{A}} h_{t-1} + \overline{\mathbf{B}} x_t, \; y_t = \overline{\mathbf{C}} h_t, \tag{2}$$

$$\overline{\mathbf{A}} = \exp(\mathbf{\Delta A}), \; \overline{\mathbf{B}} = (\mathbf{\Delta A})^{-1}(\exp(\mathbf{\Delta A}) - \mathbf{I})\mathbf{\Delta B}, \; \overline{\mathbf{C}} = \mathbf{C} \tag{3}$$

where $\mathbf{\Delta}$ represents the timescale from system states $h_{t-1}$ to the next $h_t$. The entire sequence transformation can also be represented in a convolutional form:

$$\overline{\mathbf{K}} = (\mathbf{C}\overline{\mathbf{B}}, \mathbf{C}\overline{\mathbf{A}}\overline{\mathbf{B}}, \cdots, \mathbf{C}\overline{\mathbf{A}}^{N-1}\overline{\mathbf{B}}), \; y = x * \overline{\mathbf{K}}, \tag{4}$$

where $N$ is the length of the input sequence $x$, and $\overline{\mathbf{K}} \in \mathbb{R}^N$ denotes a global convolution kernel, which can be efficiently pre-computed. However, due to the Linear Time-Invariant (LTI) nature of SSM, the parameters $(\mathbf{\Delta}, \mathbf{A}, \mathbf{B}, \mathbf{C})$ remain fixed across all time steps, which limits their ability to handle varying input sequences.

Recently, Mamba (Gu & Dao, 2023) introduced a selection mechanism that treats the parameters $(\mathbf{\Delta}, \mathbf{B}, \mathbf{C})$ as functions of the input, effectively transforming the SSM into a time-varying model:

$$h_t = \phi_{\overline{\mathbf{A}}}(x_t) h_{t-1} + \phi_{\overline{\mathbf{B}}}(x_t) x_t, \; y_t = \phi_{\overline{\mathbf{C}}}(x_t) h_t, \tag{5}$$

where $\phi_{\overline{\mathbf{A}}}(x_t)$, $\phi_{\overline{\mathbf{B}}}(x_t)$ and $\phi_{\overline{\mathbf{C}}}(x_t)$ denote the parameter matrices are dependent on the system inputs $x_t$. While the selection mechanism addresses the limitations of the LTI model, it also does not allow for parallel computation using equation 4. To tackle this challenge, Mamba introduced hardware-aware selective scanning, achieving near-linear complexity. Mamba2 (Dao & Gu, 2024) propose a refinement version of the selective SSM by leveraging structured semiseparable matrices and the state space dual framework, further enhancing performance and efficiency. In this paper, we model the relationship between the system states and system inputs based on Mamba and Mamba2, adapting it to more challenging point cloud tasks.

## 3.2 Overview

Figure 2 presents the overall framework of the DEST-based method for 3D object detection. In 3D object detection on point clouds, given a set containing $N$ points, the objective is to generate a set of 3D oriented bounding boxes with classification scores to cover all ground-truth objects. The proposed DEST-based detector primarily consists of three components: an encoder for extracting point features, a sampling module for generating the initial system states, and an ISSM-based decoder to refine system states and predict the 3D oriented bounding boxes. In this paper, we focus primarily on the decoder design, leveraging the SSM to facilitate the simultaneous updating of query points and scene points, thereby mitigating the performance limitations.

## 3.3 Interactive State Space Model

In the ISSM, we model the query points as the initial system states $h_0 \in \mathbb{R}^{K \times C}$ and the scene points as the system inputs $x \in \mathbb{R}^{M \times C}$. As shown in Figure 3, we provide the overview of the ISSM. Compared to Mamba, we expand the dimension of $\mathbf{\Delta}$ to make it state-dependent and design a spatial correlation module to generate the SSM parameters $(\mathbf{\Delta}, \mathbf{B}, \mathbf{C})$.

**Extension of $\mathbf{\Delta}$.** In Mamba, the important parameter $\mathbf{\Delta} \in \mathbb{R}^{M \times C}$ controls the balance between how much to focus on or ignore the current input. Specifically, a larger $\mathbf{\Delta}$ resets the states $h_{t-1}$ to focus on the current input $x_t$, while a smaller $\mathbf{\Delta}$ retains the states $h_{t-1}$ and disregards the current

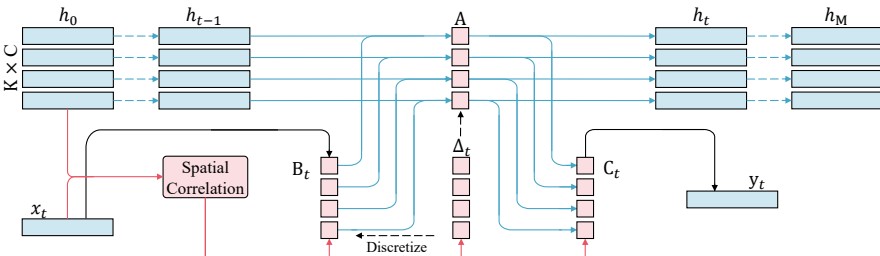

Figure 3: **Overview of the Interactive State Space Model.** In the ISSM, we model the query points as the system states and the scene points as the system inputs. We design a spatial correlation module to parameterize the SSM based on the initial system states and inputs.

input $x_t$. However, in both Mamba and Mamba2, $\boldsymbol{\Delta}$ only considers the system inputs $x$ without accounting for differences in initial system states $h_0$. In 3D object detection tasks, query points have different positions across varying scenes, and different query points focus on different system inputs. Updating the states $h$ by considering only the system inputs $x$ prevents the state points $h$ from adequately focusing on their respective regions. To address this issue, we modify $\boldsymbol{\Delta}$ to have distinct values for each system input and state, expanding $\boldsymbol{\Delta} \in \mathbb{R}^{M \times C}$ to $\boldsymbol{\Delta} \in \mathbb{R}^{M \times K \times C}$. Although the expansion of $\boldsymbol{\Delta}$ affects the discretization formula as equation 3, it does not impact the discretized state space model as equation 2. Therefore, the proposed ISSM can achieve efficient parallel computation by utilizing the hardware-aware selective scanning strategy (Gu & Dao, 2023) or the state space dual framework (Dao & Gu, 2024).

**Spatial Correlation.** The success of Mamba demonstrates that the state selection mechanism is crucial for SSMs. For 3D object detection tasks, the proposed ISSM needs to address a more complex state updating problem. A key challenge lies in enabling state points to select the appropriate scene points for self-updating. DETR-based methods (Wang et al., 2023; Shen et al., 2024) have inspired us that query points should primarily focus on points surrounding the relevant 3D bounding boxes. Therefore, we design a spatial correlation module that encodes the spatial relationships between scene points $x$ and initial state points $h_0$ to derive the parameters $(\boldsymbol{\Delta} \in \mathbb{R}^{M \times K \times C}, \mathbf{B} \in \mathbb{R}^{M \times K}, \mathbf{C} \in \mathbb{R}^{M \times K})$ in the ISSM. Specifically, for each state point $h_0^i$, we first predict a rotated 3D bounding box. We then calculate the relative offsets $\triangle P^i \in \mathbb{R}^{M \times K \times 8 \times 3}$ between the scene points and the eight vertices of the bounding box. Finally, we use an MLP to map these positional relationships to the parameters in the ISSM:

$$S^i = \sum_{j=1}^{8} \text{MLP}(\triangle P_j^i), \ \boldsymbol{\Delta_s^i} = \text{Linear}_c(S^i), \ \mathbf{B_s^i} = \text{Linear}_1(S^i), \ \mathbf{C_s^i} = \text{Linear}_1(S^i), \quad (6)$$

where $\text{Linear}_d$ is a parameterized projection to dimension d. Since there are numerous background points within the scene points, we need to prevent these from interfering with the system states. Therefore, we use the features of the scene points to modify the parameters of ISSM:

$$\boldsymbol{\Delta^i} = \text{BC}_k(\text{Linear}_c(x)) + \boldsymbol{\Delta_s^i}, \ \mathbf{B^i} = \text{BC}_k(\text{Linear}_1(x)) + \mathbf{B_s^i}, \ \mathbf{C^i} = \text{BC}_k(\text{Linear}_1(x)) + \mathbf{C_s^i}, \ (7)$$

where $\text{BC}_k$ denotes broadcasting the values to $K$ dimensions. Apart from modeling positional correlations and background information, we design an explicit delay kernel for $\boldsymbol{\Delta^i}$:

$$\boldsymbol{\Delta_g^i} = \boldsymbol{\Delta^i} \times \exp(\alpha \min(R(h_0^i) - P_x, 0)), \quad (8)$$

where $R(h_0^i)$ is the circumscribed sphere radius of the bounding box predicted by the state point $h_0^i$, $P_x$ denotes the position of the scene points and $\alpha$ is a learnable parameter. With the parameters $(\boldsymbol{\Delta_g^i}, \mathbf{B^i}, \mathbf{C^i})$, the state points $h$ can select the appropriate scene points $x$ for updating, while scene points $x$ can simultaneously acquire surrounding structural information from the state points $h$. In practice, we observe that the numerous combinations between state points $h$ and scene points $x$ result in significant memory overhead. Therefore, following VDETR (Shen et al., 2024), we utilize a smaller predefined 3D table $T \in \mathbb{R}^{10 \times 10 \times 10}$ to obtain $S^i$ of equation 6 through grid sampling. In Appendix, we further discuss the mathematical relationship between the system states in the state space model and the query points in the transformer decoder.

### 3.4 ISSM-BASED DECODER

Based on the above ISSM, we further design an ISSM-based decoder suitable for 3D point cloud detection as shown in Figure 4 (a). The ISSM-based decoder consists of four core components:

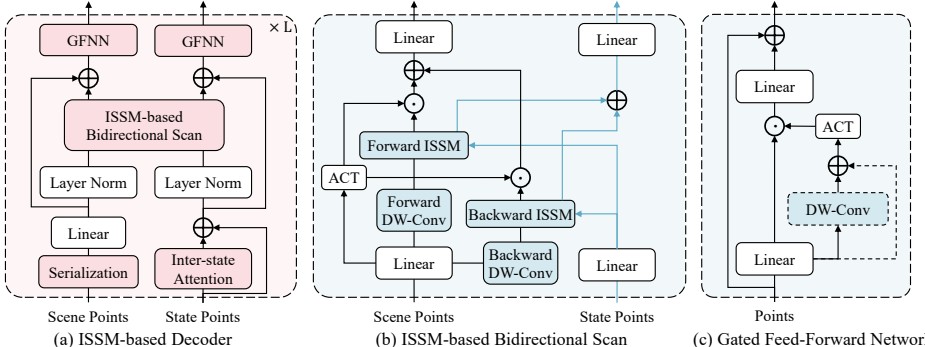

Figure 4: **(a):** Illustration of ISSM-based decoder architecture. **(b):** Detailed Structure of the ISSM-based Bidirectional Scan. **(c):** Detailed Structure of the Gated Feed-Forward Network (GFFN).

a Hilbert-based point cloud serialization strategy, an inter-state attention module, an ISSM-based Bidirectional Scan (IBS) module, and a Gated Feed-Forward Network (GFFN).

**Hilbert-based point cloud serialization strategy.** SSMs are designed for ordered 1D sequences, which are not suitable for unordered point clouds. To model the scene points as the system inputs of the ISSM, we need to serialize the scene points. Following PTv3 (Wu et al., 2024), we leverage space-filling curves to serialize point clouds. Among these space-filling curves, the Hilbert curve (Hilbert & Hilbert, 1935) is renowned for its efficient locality preservation. Thus, we generate six different serialized results for the point cloud by reordering the x, y, and z axes of the Hilbert curve, aiming for comprehensive observation of the point cloud. The detailed serialization process is shown in Appendix A.3. Additionally, unlike serialized attention in PTv3, the sequential feature modeling in ISSM is more sensitive to changes in the serialization method. Therefore, we apply different serialization methods for each decoder layer without the shuffle order strategy used in PTv3, ensuring that the ISSM-based decoder can comprehensively capture scene point features.

**Inter-state attention module.** In 3D object detection, objects in a scene often exhibit strong correlations. For example, tables and chairs commonly appear together, while beds and toilets rarely coexist in the same room. In DETR-based decoder layers (Carion et al., 2020; Misra et al., 2021; Liu et al., 2021; Shen et al., 2024), the self-attention mechanism for queries is employed to model the correlations between different objects. However, in the ISSM, there is no design specifically for interactions between state points. To capture such relationships, we employ a standard self-attention mechanism for state points. The inter-state attention module allows states to capture richer features, particularly enhancing the detection performance for objects with ambiguous boundaries or those that are challenging to distinguish from the background.

**IBS module.** To facilitate bidirectional interaction among different scene points in a single pass, we follow Vision Mamba (Zhu et al., 2024) and introduce a bidirectional scanning mechanism into our ISSM. Figure 4 (b) illustrates the detailed structure of the IBS module. Firstly, we input the forward-ordered and backward-ordered scene points into their respective forward and backward ISSMs. Both the forward and backward ISSMs use state points as system states, with each ISSM generating the updated features of the scene points and state points as outputs. We then fuse the output using a linear layer to obtain the final features of the scene points and state points. Additionally, we incorporate a depthwise convolution (Chollet, 2017) for local feature extraction of the scene points. In Appendix A.2, we provide detailed algorithmic procedures of the IBS module.

**GFFN.** The gating mechanism (Dauphin et al., 2017) introduces gated linear units, which allows the model to dynamically select activation paths based on the input. This selectivity improves the flexibility of the model, enabling the model to better capture complex patterns. Thus, we design the GFFN to replace the standard FFN, as shown in Figure 4 (c). For the ordered scene points, we also incorporate a depth-wise convolution to fully leverage the spatial structural information of the point cloud.

### 3.5 MODEL SETUPS

For a fair comparison, we build our DEST-based 3D detector based on two baseline DETR-based models: GroupFree (Liu et al., 2021) and VDETR (Shen et al., 2024), respectively. **For GroupFree baseline**, the input points $P_{in} \in \mathbb{R}^{N \times 3}$ contain only position information. We replace all trans-

former decoder layers with our proposed ISSM-based decoder layers while retaining the original spatial encodings and the original detection heads. For the training loss, we employ the same detection loss with GroupFree and introduce an additional objectness loss for the scene points. Specifically, we use an MLP head to determine whether a scene point is a foreground point and apply a binary focal loss to supervise the prediction results. **For VDETR baseline**, the input points $P_{in} \in \mathbb{R}^{N \times 6}$ include both position and RGB information. In the decoder, we also replace all transformer decoder layers with our ISSM-based decoder layers while retaining the original detection heads. Regarding the spatial encoding, we generate them for the corresponding state points using the predicted 3D bounding boxes and for the scene points using their point positions. For the training loss, we employ the same detection loss with VDETR and introduce the same objectness loss as above for the scene points. Due to space limitations, more details can be found in the Appendix.

## 4 EXPERIMENTAL RESULTS

### 4.1 DATASETS AND METRICS

**Dataset.** We evaluate our DEST-based detector on two challenging 3D indoor object detection datasets, including ScanNet V2 (Dai et al., 2017) and SUN RGB-D (Song et al., 2015). *ScanNet V2* dataset contains 1201 training samples and 312 validation samples, each annotated with per-point instance and semantic labels, as well as axis-aligned 3D bounding boxes across 18 categories. *SUN RGB-D* dataset is a monocular dataset, containing over 10000 indoor RGB-D images annotated with per-point semantic labels and oriented 3D bounding boxes across 37 categories. We follow previous methods (Qi et al., 2019; Liu et al., 2021) to evaluate our approach on the 10 most common classes of objects. The training and validation splits contain 5285 and 5050 point clouds, respectively.

**Evaluation Metrics.** Following the standard evaluation protocol (Qi et al., 2019), we evaluate our ISSM-based detector performance with the mean Average Precision ($AP_{25}$ and $AP_{50}$) under two different Intersections over Union (IoU) thresholds of 0.25 and 0.5. Since the input point clouds are obtained through random sampling, both the training and testing processes are stochastic. To ensure the reliability of the test results, we run the training 5 times and independently test each trained model 5 times. We report both the highest performance and the average results under $5 \times 5$ trials.

### 4.2 COMPARISON WITH STATE-OF-THE-ART METHODS

Different 3D detection models employ various techniques in terms of encoder architecture and bounding box parameterization, with some methods using point clouds with color information as model inputs. Therefore, it is unfair to compare the performance of different methods directly. Among these 3D detection methods, GroupFree (Liu et al., 2021) is a pioneer in designing the DETR-based method for point clouds, demonstrating strong performance across multiple datasets. VDETR (Shen et al., 2024) builds upon 3DETR (Misra et al., 2021) by introducing the positional encoding in the decoder, achieving state-of-the-art performance. To demonstrate the effectiveness of our method, we implement the DEST-based detector on top of the two DETR-based methods.

As shown in Table 1, we compare our DEST-based detector with the previous 3D object detection methods on the ScanNet V2 and SUN RGB-D datasets. The results indicate that our method significantly outperforms the baseline methods on both datasets, whether measured by the highest performance or the average results over multiple trials. For the GroupFree baseline, our DEST-based detector demonstrates substantial performance improvements across different decoder scales. For example, on the ScanNetV2 dataset, our method achieves a 4.3 increase in $AP_{50}$ based on GroupFree (S) and a 5.3 increase based on GroupFree (L). For the VDETR baseline, our DEST-based detector achieves new state-of-the-art performance on both datasets. Specifically, our approach reaches 78.8 in $AP_{25}$ and 67.9 in $AP_{50}$ on the ScanNet V2 dataset, which is 1.0 and 1.9 better than the baseline model. Additionally, our method achieves 69.2 in $AP_{25}$ and 52.2 in $AP_{50}$ on the SUN RGB-D dataset with the gains of 1.2 and 1.1. In Appendix, we further present a visual comparison of the prediction results with different baseline methods. The experimental results clearly demonstrate the effectiveness of our method. Our DEST-based decoder addresses the performance limitations caused by fixed scene point features during the query refinement process, resulting in a significant performance boost.

Table 1: **Comparison on the ScanNet V2 and SUN RGB-D datasets.** We report both the highest performance (H) and the average results (A) under multiple trials. 'RGB' indicates that the input point clouds of the methods include color information. GroupFree(S) denotes a model with a 6-layer decoder and 256 object candidates. GroupFree(L) denotes a model with a 12-layer decoder and 512 object candidates. TTA is the test-time augmentation used in VDETR.

| Method | RGB | ScanNet V2(H) $AP_{25}$ | $AP_{50}$ | ScanNet V2(A) $AP_{25}$ | $AP_{50}$ | SUN RGB-D(H) $AP_{25}$ | $AP_{50}$ | SUN RGB-D(A) $AP_{25}$ | $AP_{50}$ |
|---|---|---|---|---|---|---|---|---|---|
| VoteNet (Qi et al., 2019) | ✗ | 62.9 | 39.9 | - | - | 57.7 | - | - | - |
| HGNet (Chen et al., 2020) | ✗ | 61.3 | 34.4 | - | - | 61.6 | - | - | - |
| 3D-MPA (Engelmann et al., 2020) | ✗ | 64.2 | 49.2 | - | - | - | - | - | - |
| MLCVNet (Xie et al., 2020) | ✗ | 64.5 | 41.4 | - | - | 59.8 | - | - | - |
| GSDN (Gwak et al., 2020) | ✗ | 62.8 | 34.8 | - | - | - | - | - | - |
| H3DNet (Zhang et al., 2020) | ✗ | 64.4 | 43.4 | - | - | 60.1 | 39.0 | - | - |
| BRNet (Cheng et al., 2021) | ✗ | 66.1 | 50.9 | - | - | 61.1 | 43.7 | - | - |
| 3DETR (Misra et al., 2021) | ✗ | 65.0 | 47.0 | - | - | 59.1 | 32.7 | - | - |
| VENet (Xie et al., 2021) | ✗ | 67.7 | - | - | - | 62.5 | 39.2 | - | - |
| GroupFree(S)(Liu et al., 2021) | ✗ | 67.3 | 48.9 | 66.3 | 48.5 | 63.0 | 45.2 | 62.6 | 44.4 |
| GroupFree(L)(Liu et al., 2021) | ✗ | 69.1 | 52.8 | 68.6 | 51.8 | - | - | - | - |
| RBGNet (Wang et al., 2022b) | ✗ | 70.6 | 55.2 | 69.9 | 54.7 | 64.1 | 47.2 | 63.6 | 46.3 |
| HyperDet3D (Zheng et al., 2022) | ✗ | 70.9 | 57.2 | - | - | 63.5 | 47.3 | - | - |
| LeadNet (Wang et al., 2023) | ✗ | 68.0 | 51.3 | - | - | 63.4 | 45.8 | - | - |
| FCAF3D (Rukhovich et al., 2022) | ✓ | 71.5 | 57.3 | 70.7 | 56.0 | 64.2 | 48.9 | 63.8 | 48.2 |
| TR3D (Rukhovich et al., 2023) | ✓ | 72.9 | 59.3 | 72.0 | 57.4 | 67.1 | 50.4 | 66.3 | 49.6 |
| CAGroup3D (Wang et al., 2022a) | ✓ | 75.1 | 61.3 | 74.5 | 60.3 | 66.8 | 50.2 | 66.4 | 49.5 |
| VDETR (Shen et al., 2024) | ✓ | 77.4 | 65.0 | 76.8 | 64.5 | 67.5 | 50.4 | 66.8 | 49.7 |
| VDETR(TTA) (Shen et al., 2024) | ✓ | 77.8 | 66.0 | 77.0 | 65.3 | 68.0 | 51.1 | 67.5 | 50.0 |
| GroupFree(S)(Liu et al., 2021) | ✗ | 67.3 | 48.9 | 66.3 | 48.5 | 63.0 | 45.2 | 62.6 | 44.4 |
| + DEST(ours) | ✗ | 68.8(+1.5) | 53.2(+4.3) | 67.9(+1.6) | 52.7(+4.2) | 65.3(+2.3) | 48.4(+3.2) | 64.7(+2.1) | 47.6(+3.2) |
| GroupFree(L)(Liu et al., 2021) | ✗ | 69.1 | 52.8 | 68.6 | 51.8 | - | - | - | - |
| + DEST(ours) | ✗ | 71.3(+2.2) | 58.1(+5.3) | 70.5(+1.9) | 56.8(+5.0) | - | - | - | - |
| VDETR (Shen et al., 2024) | ✓ | 77.4 | 65.0 | 76.8 | 64.5 | 67.5 | 50.4 | 66.8 | 49.7 |
| + DEST(ours) | ✓ | 78.5(+1.1) | 66.6(+1.6) | 77.8(+1.0) | 66.2(+1.7) | 68.4(+0.9) | 51.8(+1.4) | 67.4(+0.8) | 50.9(+1.2) |
| VDETR(TTA) (Shen et al., 2024) | ✓ | 77.8 | 66.0 | 77.0 | 65.3 | 68.0 | 51.1 | 67.5 | 50.0 |
| + DEST(ours) | ✓ | 78.8(+1.0) | 67.9(+1.9) | 78.3(+1.3) | 66.9(+1.6) | 69.2(+1.2) | 52.2(+1.1) | 68.8(+1.3) | 51.6(+1.6) |

## 4.3 ABLATION EXPERIMENTS

In this section, we verify the key design modules of our DEST-based detector. All ablation experiments are conducted on the ScanNet V2 dataset with the GroupFree(S) baseline. Following GroupFree (Liu et al., 2021), we report the average performance of 25 trials by default.

**Effect of the designed modules.** As shown in Table 2, we incrementally add each designed module in an SSM-based baseline. The SSM-based baseline is also built on GroupFree(S), using the standard Mamba2 block and FFN as the decoder components. The SSM-based baseline lacks the adaptability to the state points and cannot handle unordered point cloud inputs, resulting in a significant performance drop. Introducing the serialization and bidirectional scan strategies improves the model performance, but still lags behind GroupFree(S). A substantial performance boost is observed when the proposed ISSM replaces the Mamba2 block. This boost is attributed to the adaptive ability of the ISSM, allowing state points to select appropriate scene points for feature updating. Further incorporating an inter-state mechanism and a gated linear unit achieves the best performance.

**Effect of the ISSM.** ISSM is the core of the proposed DEST and is responsible for the information exchange and updating between scene and state points. Compared with the selective SSM, ISSM extends the dimension of $\Delta$ and introduces a spatial correlation module and a delay kernel to assist scene points in generating the SSM parameters. To analyze the impact of parameter generation, we evaluate different combinations of parameter generation methods. As shown in the second and third rows of Table 3, using the spatial correlation module and delay kernel to generate SSM parameters leads to a significant performance improvement. The results indicate that allowing state points to focus on their relevant regions is crucial for designing SSMs for 3D object detection. Moreover,

Table 2: **Effect of the designed modules.** We progressively add the proposed modules to the SSM-based baseline to verify the contribution of each module.

| Method | $AP_{25}$ | $AP_{50}$ |
|---|---|---|
| GroupFree(S) | 66.3 | 48.5 |
| Baseline | 60.2 | 41.6 |
| w/ serialization | 62.8 | 43.5 |
| w/ bidirectional scan | 63.7 | 44.8 |
| w/ ISSM | 67.1 | 50.6 |
| w/ inter-state attention | 67.6 | 51.9 |
| w/ GFFN | **67.9** | **52.7** |

Table 3: **Effect of the ISSM.** Here, $\phi(x)$ represents generating the SSM parameters using the scene points $x$. $\phi(S)$ indicates the incorporation of spatial correlation $S$ to generate the parameters. $R(h)$ denotes the addition of a delay kernel for $\mathbf{\Delta}$.

| $\phi(x)$ | $\phi(S)$ | $R(h)$ | $AP_{25}$ | $AP_{50}$ |
|---|---|---|---|---|
| ✓ | ✗ | ✗ | 64.3 | 46.1 |
| ✓ | ✓ | ✗ | 67.2 | 51.3 |
| ✓ | ✗ | ✓ | 67.4 | 50.8 |
| ✗ | ✓ | ✓ | 66.7 | 49.4 |
| ✓ | ✓ | ✓ | **67.9** | **52.7** |

Table 4: **Effect of the simultaneous updating.** "Fixed" denotes that the scene point features used in each decoder layer remain as those outputted by the encoder. "GroupFree$^*$" indicates that we introduced a GFFN in each decoder layer to update the scene point features.

| Method | Fixed | $AP_{25}$ | $AP_{50}$ |
|---|---|---|---|
| GroupFree(S) | ✓ | 66.3 | 48.5 |
| GroupFree$^*$ | ✗ | 66.8 | 49.0 |
| DEST(ours) | ✓ | 66.6 | 49.3 |
| DEST(ours) | ✗ | 67.9 | 52.7 |

Table 5: **Comparison of the model parameters and inference speed on ScanNet V2.** "Param." denotes the total parameters.

| Method | Param. | Latency | $AP_{25}$ | $AP_{50}$ |
|---|---|---|---|---|
| GroupFree(S) | 13.8 M | 21 ms | 67.3 | 48.9 |
| + DEST(ours) | 19.6 M | 34 ms | 68.8 | 53.2 |
| GroupFree(L) | 28.2 M | 68 ms | 69.1 | 52.8 |
| + DEST(ours) | 39.8 M | 98 ms | 71.3 | 58.1 |
| FCAF3D | 67.2 M | 138 ms | 71.5 | 57.3 |
| CAGroup3D | 120.7 M | 472 ms | 75.1 | 61.3 |
| VDETR | 75.6 M | 238 ms | 77.8 | 66.0 |
| + DEST(ours) | 80.1 M | 263 ms | 78.8 | 67.9 |

comparing the fourth and fifth rows, we find that considering only spatial relationships also results in a drop in performance. When the above modules are combined to generate SSM parameters, the DEST-based detector achieves the best performance.

**Effect of the scene point updating.** To evaluate the impact of scene point feature updating, we conduct ablation experiments as shown in Table 4. In the second row, we add the proposed GFFN to GroupFree(S) to update the scene point features. However, due to its limited receptive field, it only achieves a slight performance improvement. In the third row, we fix the scene point features in the proposed DEST-based detector, which results in a significant performance decline. These results demonstrate the effectiveness of the simultaneous updating in DEST-based methods. In the DEST-based methods, scene points can capture global contextual information through state points for self-updating, thereby providing more effective information for subsequent decoder layers.

## 4.4 COMPARISON OF PARAMETERS AND INFERENCE SPEED

The model complexity of both DETR-based and DEST-based methods is determined by the number of scene points and query points. To ensure a fair comparison, we compare the model parameters and inference speed with different DETR-based baseline methods. All experiments are conducted on a Tesla V100 GPU. As shown in Table 5, our method demonstrates comparable model parameter efficiency and inference speed while enabling simultaneous updates of scene points and state points. Although there is a slight increase in parameter count and computational cost compared to the baseline methods, this overhead is acceptable given the substantial performance gains.

## 5 CONCLUSION

In this paper, we identify a crucial issue in DETR-based decoders: the fixed scene point features lead to suboptimal refinement of query points in the later layers. To address this issue, we introduce a novel DEST-based method that simultaneously updates scene and query point features. The key contribution lies in designing the ISSM, which models query points as system states and scene points as system inputs, allowing simultaneous updates with linear complexity. Our DEST-based method demonstrates significant advantages through comparison experiments with two DETR-based methods, and comprehensive ablation studies validate the effectiveness of each designed module.

## 6    ACKNOWLEDGEMENT

This work was supported by Hainan Province Science and Technology Special Fund (Grant No. ATIC-2023010004), National Natural Science Foundation of China (No. 12150007, No. 62071122), Basic Strengthening Program Laboratory Fund (No. NKLDSE2023A009), and Youth Innovation Promotion Association of CAS.

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

# A APPENDIX

## A.1 RELATIONSHIP BETWEEN QUERY AND STATE

In this section, we further analyze the relationship between queries in the attention mechanism and the states in the State Space Model (SSM). For 3D object detection, the transformer decoder employs an attention mechanism to facilitate the interaction between object candidate point features $h_0 \in \mathbb{R}^{K \times C}$ and scene point features $x \in \mathbb{R}^{M \times C}$. Specifically, the object candidate point features $h_0$ serve as the query $Q_0$, while the scene point features $x$ act as the key $K$ and value $V$. The attention mechanism is then used to update $h_0$:

$$Q_m^i = \frac{\sum_{j=1}^{m} \text{sim}(Q_0^i, K_j) V_j}{\sum_{j=1}^{m} \text{sim}(Q_0^i, K_j)}, \tag{9}$$

where $\text{sim}(,)$ denotes the feature similarity calculation, $Q_m^i$ represents the $i$-th query updated using the first $m$ scene point features, $Q_0^i$ denotes the initial $i$-th query, and $K$ and $V$ correspond to the $j$-th key and value, respectively. The above equation can be reformulated as follows:

$$
\begin{aligned}
Q_m^i &= \frac{[\sum_{j=1}^{m-1} \text{sim}(Q_0^i, K_j) V_j] + [\text{sim}(Q_0^i, K_m) V_m]}{\sum_{j=1}^{m} \text{sim}(Q_0^i, K_j)} \\
&= \frac{\sum_{j=1}^{m-1} \text{sim}(Q_0^i, K_j) V_j}{\sum_{j=1}^{m} \text{sim}(Q_0^i, K_j)} + \frac{\text{sim}(Q_0^i, K_m)}{\sum_{j=1}^{m} \text{sim}(Q_0^i, K_j)} V_m \\
&= \frac{\sum_{j=1}^{m-1} \text{sim}(Q_0^i, K_j)}{\sum_{j=1}^{m} \text{sim}(Q_0^i, K_j)} \frac{\sum_{j=1}^{m-1} \text{sim}(Q_0^i, K_j) V_j}{\sum_{j=1}^{m-1} \text{sim}(Q_0^i, K_j)} + \frac{\text{sim}(Q_0^i, K_m)}{\sum_{j=1}^{m} \text{sim}(Q_0^i, K_j)} V_m \\
&= \frac{\sum_{j=1}^{m-1} \text{sim}(Q_0^i, K_j)}{\sum_{j=1}^{m} \text{sim}(Q_0^i, K_j)} Q_{m-1}^i + \frac{\text{sim}(Q_0^i, K_m)}{\sum_{j=1}^{m} \text{sim}(Q_0^i, K_j)} V_m.
\end{aligned}
\tag{10}
$$

Based on the above equation, the query feature $Q_m^i$ updated with the first $m$ points can be regarded as a linear combination of the query feature $Q_{m-1}^i$ updated with the first $m-1$ points and the feature $V_m$ of the $m$-th scene point. Furthermore, we rewrite the equation 10 in the following form:

$$Q_m^i = A_m^i Q_{m-1}^i + B_m^i V_m, A_m^i = \frac{\sum_{j=1}^{m-1} \text{sim}(Q_0^i, K_j)}{\sum_{j=1}^{m} \text{sim}(Q_0^i, K_j)}, B_m^i = \frac{\text{sim}(Q_0^i, K_m)}{\sum_{j=1}^{m} \text{sim}(Q_0^i, K_j)}. \tag{11}$$

In the discrete SSM, we observe that the interaction process between system states and scene points shares a similar form with the equation 11:

$$h_m^i = \overline{A}_m^i h_{m-1}^i + \overline{B}_m^i x_m, \tag{12}$$

where $h_m^i$ denotes the $i$-th system state updated using the first $m$ scene points, $x_m$ denotes the $m$-th input of scene point features, while $\overline{A}_m^i$ and $\overline{B}_m^i$ are the parameters of the SSM. Therefore, we consider the attention mechanism in the transformer decoder to be a type of SSM, which designs the query update strategy through the feature similarity $\text{sim}(,)$. The previous SSMs only utilize $x_m$ to generate the SSM parameters $\overline{A}_m^i$ and $\overline{B}_m^i$, making them unsuitable for modeling query points in 3D object detection tasks. However, the update strategy based on feature similarity $\text{sim}(,)$ requires significant computational overhead. Consequently, in the proposed ISSM, we utilize a spatial correlation module and scene point features to generate SSM parameters as equation 7 and equation 8, reducing the model overhead without compromising performance. Based on the above analysis, we believe that the system states in our ISSM can effectively serve the role of the queries with the appropriate parameters $\overline{A}_m^i$ and $\overline{B}_m^i$.

## A.2 ISSM-BASED BIDIRECTIONAL SCAN MODULE

In the ISSM-based decoder, the core design is the ISSM-based Bidirectional Scan (IBS) module. The IBS module integrates an interactive state space model with bidirectional sequential modeling tailored for point clouds. To illustrate the process of the IBS module more clearly, we provide the

---

**Algorithm 1** ISSM-based Bidirectional Scan Module Process

---

**Require:** scene points $x$: (B, M, C), initial state points $h_0$: (B, K, C)
**Ensure:** updated scene points $y$: (B, M, C), final state points $h_M$: (B, K, C)
 1: /* normalize the input scene points $x$ */
 2: $x'$: (B, M, C) $\leftarrow$ Norm$(x)$, $h'_0$: (B, K, C) $\leftarrow$ Norm$(h_0)$
 3: $\hat{x}$: (B, M, E) $\leftarrow$ Linear$^x(x')$, $z$: (B, M, E) $\leftarrow$ Linear$^z(x')$
 4: $\hat{h_0}$: (B, K, E) $\leftarrow$ Linear$^h(h'_0)$
 5: /* process with different direction */
 6: **for** $o$ in $\{$forward, backward$\}$ **do**
 7:    $\hat{x}_o$: (B, M, E) $\leftarrow$ SiLU(DW_Conv1d$_o(\hat{x})$)
 8:    /* $S$ is the spatial correlation of scene points and state points: (B, M, K, D) */
 9:    $\mathbf{B}_o$: (B, M, K) $\leftarrow$ Broadcast$_k$(Linear$_1^{\mathbf{B},o}(\hat{x}_o)$) + Linear$_1^{\mathbf{B},s}(S)$
10:    $\mathbf{C}_o$: (B, M, K) $\leftarrow$ Broadcast$_k$(Linear$_1^{\mathbf{C},o}(\hat{x}_o)$) + Linear$_1^{\mathbf{C},s}(S)$
11:    /* softplus ensures positive $\mathbf{\Delta}_o$, $\mathbf{\Delta}_{delay}$ is the proposed delay kernel: (B, M, K) */
12:    $\mathbf{\Delta}_o$: (B, M, K, E) $\leftarrow$ log$(1 + $exp(Broadcast$_k$(Linear$_E^{\mathbf{\Delta},o}(\hat{x}_o)$) + Linear$_E^{\mathbf{\Delta},s}(S)$)) \otimes \mathbf{\Delta}_{delay}$
13:    /* Parameter$_o^{\mathbf{A}}$ is learnable parameter: (M, E) */
14:    $\overline{\mathbf{A}}_o$: (B, M, K, E) $\leftarrow$ $\mathbf{\Delta}_o \otimes$ Parameter$_o^{\mathbf{A}}$
15:    $\overline{\mathbf{B}}_o$: (B, M, K, E) $\leftarrow$ $\mathbf{\Delta}_o \otimes \mathbf{B}_o$
16:    $\hat{y}_o$: (B, M, E), $\hat{h}_N^o$: (B, K, E) $\leftarrow$ **SSM**$(\overline{\mathbf{A}}_o, \overline{\mathbf{B}}_o, \mathbf{C}_o)(\hat{x}_o, \hat{h_0})$
17: **end for**
18: /* gated linear unit and residual connection */
19: $y'_{\text{forward}}$: (B, M, E) $\leftarrow$ $\hat{y}_{\text{forward}} \odot$ SiLU$(z)$
20: $y'_{\text{backward}}$: (B, M, E) $\leftarrow$ $\hat{y}_{\text{backward}} \odot$ SiLU$(z)$
21: $y$: (B, M, C) $\leftarrow$ Linear$^y(y'_{\text{forward}} + y'_{\text{backward}}) + x$
22: $h_M$: (B, K, C) $\leftarrow$ Linear$^h(\hat{h}_N^{\text{forward}} + \hat{h}_N^{\text{backward}}) + h_0$
23: **return** $y$ and $h_M$

---

detailed operations of the IBS module in Algorithm 1. The input scene points $x$ and initial state points $h_0$ are first normalized by the normalization layer. Next, we linearly project the normalized features $x'$ to $\hat{x}$ and $z$, and the $h'_0$ to $\hat{h_0}$. Before we process the $\hat{x}$ from the forward and backward directions, we calculate the spatial correlation $S$ of scene points and state points as equation 6 and the explicit delay kernel $\mathbf{\Delta}_{delay}$ as equation 8. For each direction, we first apply the depthwise convolution to the $\hat{x}$ and get the $\hat{x}_o$. The SSM parameters $(\mathbf{B}_o, \mathbf{C}_o, \mathbf{\Delta}_o)$ are generated based on the $\hat{x}_o$, $S$ and $\mathbf{\Delta}_{delay}$. Then we use the $\mathbf{\Delta}_o$ to transform the $\overline{\mathbf{A}}_o, \overline{\mathbf{B}}_o$. Finally, we get the updated scene points $\hat{y}_o$ and updated state points $\hat{h}_N^o$ through the SSM. For the scene points $\hat{y}_{\text{forward}}, \hat{y}_{\text{backward}}$, we gated them by the $z$ and add them together. Similarly, for the state points $\hat{h}_N^{\text{forward}}, \hat{h}_N^{\text{backward}}$, we add them directly. The updated scene and state points are then connected to $x$ and $h_0$ using residual connections, respectively, to produce the final outputs $y$ and $h_M$.

### A.3 POINT CLOUD SERIALIZATION

To enable unordered point clouds to be inputted into the SSM in an ordered manner, we use Hilbert space-filling curves to sort the scene points. With the different priority of the x, y, and z axes, we can get six different space-filling curves, which we denote as "xyz", "xzy", "yxz", "yzx", "zxy", and "zyx". To better illustrate the point cloud serialization process, we visualize the 2D Hilbert space-filling curves and the serialization results of the 2D point set in Figure 5. Specifically, we divide the space into uniformly sized grids and then sort each grid based on the space-filling curve. Finally, we sort the point set according to the indices of the grids where the points are located. As shown in Figure 5, different space-filling curves yield varying ordered results for the point sets. These different serialization results represent observations of point clouds from different perspectives. Therefore, we utilize six space-filling curves in 3D space, employing different serialization methods at different decoder layers.

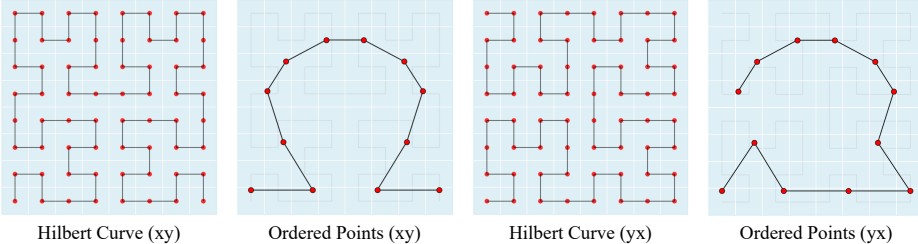

| Hilbert Curve (xy) | Ordered Points (xy) | Hilbert Curve (yx) | Ordered Points (yx) |

Figure 5: **Visualization of Hilbert space-filling curves and Hilbert-based point cloud serialization.** To facilitate analysis, we visualize all images in 2D space. By swapping the priority of the coordinate axes, we can obtain different Hilbert space-filling curves. Different spatial filling curves can yield distinct serialization results for the same point set.

### A.4 MORE IMPLEMENTATION DETAILS

#### A.4.1 MODEL SETUP DETAILS

**GroupFree baseline.** Given the input points $P_{in} \in \mathbb{R}^{N \times 3}$ without RGB information, we employ PointNet++ (Qi et al., 2017b) as the point cloud encoder for a fair comparison. In addition, we utilize the same initial object candidate sampling module as employed in GroupFree (Liu et al., 2021). However, unlike GroupFree, where the object candidate points are used as the initial query points, they are treated as the initial state points in our ISSM-based model. In the decoder, we replace all transformer decoder layers with our proposed ISSM-based decoder layers while retaining the original spatial encoding and the original detection heads.

**VDETR baseline.** Given the input points $P_{in} \in \mathbb{R}^{N \times 6}$ with RGB information, we employ the same 3D sparse convolution network of VDETR (Shen et al., 2024) as the point cloud encoder for a fair comparison. Additionally, we utilize the same initial object query sampling module as employed in VDETR (Shen et al., 2024) to sample the initial state points. In the decoder, we also replace all transformer decoder layers with our proposed ISSM-based decoder layers while retaining the original detection heads. Regarding the spatial encoding, we generate them for the corresponding state points using the predicted 3D bounding boxes and for the scene points using their point positions.

#### A.4.2 TRAINING DETAILS

**Training with GroupFree baseline.**

*For the ScanNet V2 dataset*, we use 50k points as input. In the training phase, we adopt the same data augmentation as in GroupFree (Liu et al., 2021), including random flip, random rotation along the z-axis $[-5°, 5°]$, and random scaling $[0.9, 1.1]$. The encoder consists of four set abstraction layers and two feature propagation layers. After feature extraction by the encoder, a total of 1024 points are output as scene points. The detector is trained from scratch using the AdamW (Loshchilov, 2017) optimizer ($\beta_1 = 0.9, \beta_2 = 0.999$) for 400 epochs. The weight decay is set to 5e-4. The initial learning rate is 6e-3, which decays by a factor of 10 at the 280th and 340th epochs. The learning rate of the decoder is set as $\frac{1}{10}$ of that in the encoder.

*For the SUN RGB-D dataset*, we use 20k points as input. The encoder architecture and data augmentation are the same as those used for ScanNet V2. Following GroupFree, we include an additional orientation prediction branch in all decoder layers. During training, the detector is trained from scratch using the AdamW optimizer ($\beta_1 = 0.9, \beta_2 = 0.999$) with 600 epochs. The weight decay is set to 1e-7. The initial learning rate is 4e-3, which decays by a factor of 10 at the 420th epoch, the 480th epoch, and the 540th epoch. The learning rate of the decoder is set as $\frac{1}{20}$ of that in the encoder.

**Training with VDETR baseline.**

Table 6: **Effect of the serialization strategy.** "xyz"represents the axis priority order of the Hilbert space-filling curve.

| Strategy | $AP_{25}$ | $AP_{50}$ |
|---|---|---|
| {"xyz"} $\times$ 6 | 66.6 | 51.1 |
| {"xyz", "xzy"} $\times$ 3 | 67.1 | 51.7 |
| {"xyz", "yzx", "zxy"} $\times$ 2 | 67.4 | 52.1 |
| {"xyz", "xzy", "yxz", "yzx", "zxy", "zyx"} | 67.9 | 52.7 |

Table 7: **Effect of the bidirectional scan.** "Uni-Scan" denotes using the unidirectional scan while "Bi-Scan" denotes using the bidirectional scan. "Channel Flip" refers to reversing along the feature channels.

| Strategy | $AP_{25}$ | $AP_{50}$ |
|---|---|---|
| Uni-Scan | 67.1 | 50.9 |
| Bi-Scan | 67.9 | 52.7 |
| Bi-Scan w/ Channel Flip | 67.7 | 52.4 |

*For the ScanNet V2 dataset*, we use 100k points as input. In the training phase, we adopt the data augmentation, including random cropping, random sampling, random flipping, random rotation along the z-axis $[-5°, 5°]$, random translation $[-0.4, 0.4]$, and random scaling $[0.6, 1.4]$. After feature extraction by the encoder, a total of 4096 points are output as scene points. The detector is trained from scratch using the AdamW optimizer ($\beta_1 = 0.9, \beta_2 = 0.999$) for 540 epochs. The weight decay is set to 0.1. The initial learning rate is 7e-4, which is warmed up for 9 epochs and then is dropped to 1e-6 using the cosine schedule during the entire training process.

*For the SUN RGB-D dataset*, we use 100k points as input. The encoder architecture, data augmentation and training settings are the same as those used for ScanNet V2. In the decoder, we include an additional orientation prediction branch in all decoder layers.

## A.5 MORE ABLATION EXPERIMENTS

In this section, we provide additional ablation experiments focusing on several modules designed in our model: the serialization strategy, bidirectional scanning strategy, gated feed-forward network, and depth-wise convolution. All ablation experiments are conducted on the ScanNet V2 dataset with the GroupFree(S) baseline, and we report the average performance of 25 trials by default.

**Effect of the serialization strategy.** We conduct ablation experiments on the serialization strategy used in the ISSM-based decoder. As shown in Table 6, employing a single space-filling curve for serialization leads to a decrease in model performance. This is because a single serialization method ensures that adjacent points in the sequence are spatially adjacent, but it cannot guarantee that all spatially adjacent points remain adjacent in the sequence. Observing from multiple perspectives is essential for better modeling the local relationships of high-dimensional data. The experimental results also demonstrate that using different types of space-filling curves at different layers achieves the best performance.

**Effect of the bidirectional scan.** To validate the effect of the bidirectional scanning on the proposed ISSM-based method, we conduct ablation experiments as shown in Table 7. It is evident that bidirectional scanning significantly improves model performance compared to unidirectional scanning. In unidirectional scanning, scene points later in the sequence can access information from earlier points, but earlier points cannot access information from later ones. The bidirectional scanning strategy enables bidirectional interaction between scene points, leading to enhanced performance. Additionally, we consider using a channel flip strategy, which enhances inter-channel correlations by reversing feature channels. However, the channel flip strategy is unsuitable for our proposed method, resulting in a slight performance decrease.

**Ablation on the model size.** For 3D object detection models, the feature dimension, decoder depth, and the number of initial object candidates all significantly impact model performance. To design

Table 8: **Evaluation on different model size.** "Encoder width" represents the feature dimension of the encoder output, "# of layers" denotes the decoder depth, and "# of object candidates" denotes the number of system states in the ISSM-based decoder.

| Encoder width | # of layers | # of object candidates | $AP_{25}$ | $AP_{50}$ |
|---|---|---|---|---|
| 288 | 2 | 256 | 66.2 | 45.9 |
| 288 | 4 | 256 | 67.1 | 49.3 |
| 288 | 6 | 256 | 67.9 | 52.7 |
| 288 | 8 | 256 | 68.3 | 54.1 |
| 288 | 12 | 256 | 68.6 | 54.8 |
| 288 | 6 | 256 | 67.9 | 52.7 |
| 288 | 6 | 512 | 68.7 | 53.6 |
| 288 | 6 | 1024 | 68.8 | 53.3 |
| 288 | 6 | 256 | 67.9 | 52.7 |
| 576 | 6 | 256 | 68.4 | 54.2 |
| 576 | 12 | 256 | 69.8 | 56.5 |
| 576 | 12 | 512 | 70.5 | 56.8 |

Table 9: **Effect of the GFFN.** "GLU on $x$" denotes using the gated linear unit on scene points $x$ while "GLU on $h$" denotes using the gated linear unit on state points $h$.

| GLU on $x$ | GLU on $h$ | $AP_{25}$ | $AP_{50}$ |
|---|---|---|---|
| ✗ | ✗ | 67.6 | 51.9 |
| ✗ | ✓ | 67.8 | 52.5 |
| ✓ | ✗ | 67.6 | 52.3 |
| ✓ | ✓ | **67.9** | **52.7** |

a model with better performance under a smaller model overhead, we conduct ablation experiments on model size. As shown in Table 8, we experiment with different combinations of encoder width, decoder depth, and the number of object candidates. First, increasing the number of decoder layers leads to a continuous and significant performance improvement, particularly in $AP_{50}$. This result demonstrates that the ISSM-based decoder effectively updates the features of scene points, aiding the query points in more accurately locating object positions. Second, moderately increasing the number of initial object candidates also enhances model performance. However, when the number becomes too large, numerous background points are introduced as negative samples, reducing detection accuracy. Lastly, increasing the encoder feature dimension provides the model with richer information, thereby improving performance. By selecting a larger encoder width, a deeper decoder, and an appropriate number of query points, our model achieves optimal performance.

**Effect of the GFFN.** Introducing a gated linear unit (GLU) in the feed-forward network aims to enhance feature modeling for both scene points and state points, allowing the model to selectively pass or suppress certain features. This dynamic gating mechanism strengthens the ability to capture complex patterns, improves training efficiency, and makes it more effective in handling long-range dependencies. As shown in Table 9, including the gated linear unit positively impacts model performance. Applying the GLU on both scene and state points leads to the best detection performance.

**Effect of the local feature aggregation.** In the ISSM-based decoder layer, we employ two depth-wise convolutions in ISSM and GFFN for local feature aggregation of scene points. To analyze the impact of this local feature aggregation on model performance, we conduct the ablation experiments shown in Table 10. Eliminating the depth-wise convolutions from the ISSM-based decoder layer results in a significant decrease in model performance. Furthermore, adding depth-wise convolutions to the ISSM is more effective in improving detection accuracy than adding them to the GFFN. We also analyze the impact of the kernel size in depth-wise convolutions on model performance. We find that when the kernel size reaches 8, the model performance does not continue to increase with larger kernel sizes. Therefore, in our ISSM-based decoder, we select a kernel size of 8 to balance performance and computational cost.

Table 10: **Effect of the local feature aggregation.** "Kernel" denotes the kernel size in depth-wise convolution.

| in ISSM | in GFFN | Kernel | $AP_{25}$ | $AP_{50}$ |
|---|---|---|---|---|
| ✗ | ✗ | / | 66.8 | 51.8 |
| ✓ | ✗ | 8 | 67.6 | 52.4 |
| ✗ | ✓ | 8 | 67.2 | 52.3 |
| ✓ | ✓ | 8 | 67.9 | 52.7 |
| ✓ | ✓ | 4 | 67.4 | 52.2 |
| ✓ | ✓ | 12 | 68.1 | 52.6 |
| ✓ | ✓ | 16 | 67.8 | 52.6 |

Table 11: **Comparison of different scene point feature update methods.**

| Method | $AP_{25}$ | $AP_{50}$ |
|---|---|---|
| GroupFree | 66.3 | 48.5 |
| w standard Transformer | 66.7 | 49.1 |
| w FlashAttention | 66.5 | 49.2 |
| w selective SSM | 67.0 | 50.1 |
| DEST (Ours) | 67.9 | 52.7 |

## A.6 VISUAL COMPARISON

In this section, we present a visual comparison of the prediction results of our method and DETR-based methods. To ensure a fair comparison, we apply the same post-processing steps to the detection results. First, we filter out low-quality predictions using a class confidence threshold of 0.3, followed by class-specific Non-Maximum Suppression with an IoU threshold of 0.5 to remove redundant bounding boxes. As shown in Figure 6, Figure 7, and Figure 8, we first compare the prediction results of our method with three baseline methods on the ScanNet V2 dataset. Our method successfully detects most target objects across various scenes, with the predicted bounding boxes being more closely aligned with the ground truth labels. Subsequently, we present a visual comparison of the prediction results on the SUN RGB-D dataset. Since VDETR does not provide an official implementation for the SUN RGB-D dataset, we only include a visual comparison with the GroupFree baseline as shown in Figure 9. Since VDETR does not provide an official implementation for the SUN RGB-D dataset, we only include a visual comparison with the GroupFree baseline. Unlike the ScanNet dataset, the SUN RGB-D dataset is generated from single images, which results in point cloud data with more pronounced occlusions and uneven density, making the detection task more challenging. Nevertheless, our method still achieves visually satisfactory detection results.

## A.7 DISCUSSIONS ABOUT THE CHOICE FOR SSM

In this subsection, we discuss the choice of SSM for the DEST framework. To address the performance bottleneck caused by fixed scene point features in transformer decoders, a straightforward approach is to introduce a module in the decoder to update scene point features. To validate the effectiveness of SSM in updating scene point features, we provide numerical comparisons with standard self-attention, FlashAttention (Dao et al., 2022), and selective SSM (Gu & Dao, 2023). As shown in Table 11, attention-based methods improve the detection accuracy of the baseline model but still fall short compared to methods employing SSMs. This limitation arises because global attention is less effective at handling local relationships in point clouds and may introduce noise that disrupts scene point feature updates. In contrast, selective SSM uses 1D convolution to model local relationships and incorporates a hidden state selection mechanism to focus on extracting relevant features. Therefore, we adopt SSM to address the issue of fixed scene point features in the decoder. In our proposed DEST method, query points representing foreground objects serve as state points, providing richer and more task-specific foreground information to enhance scene point feature updates.

Ground-truth          GroupFree (S)          DEST (ours)

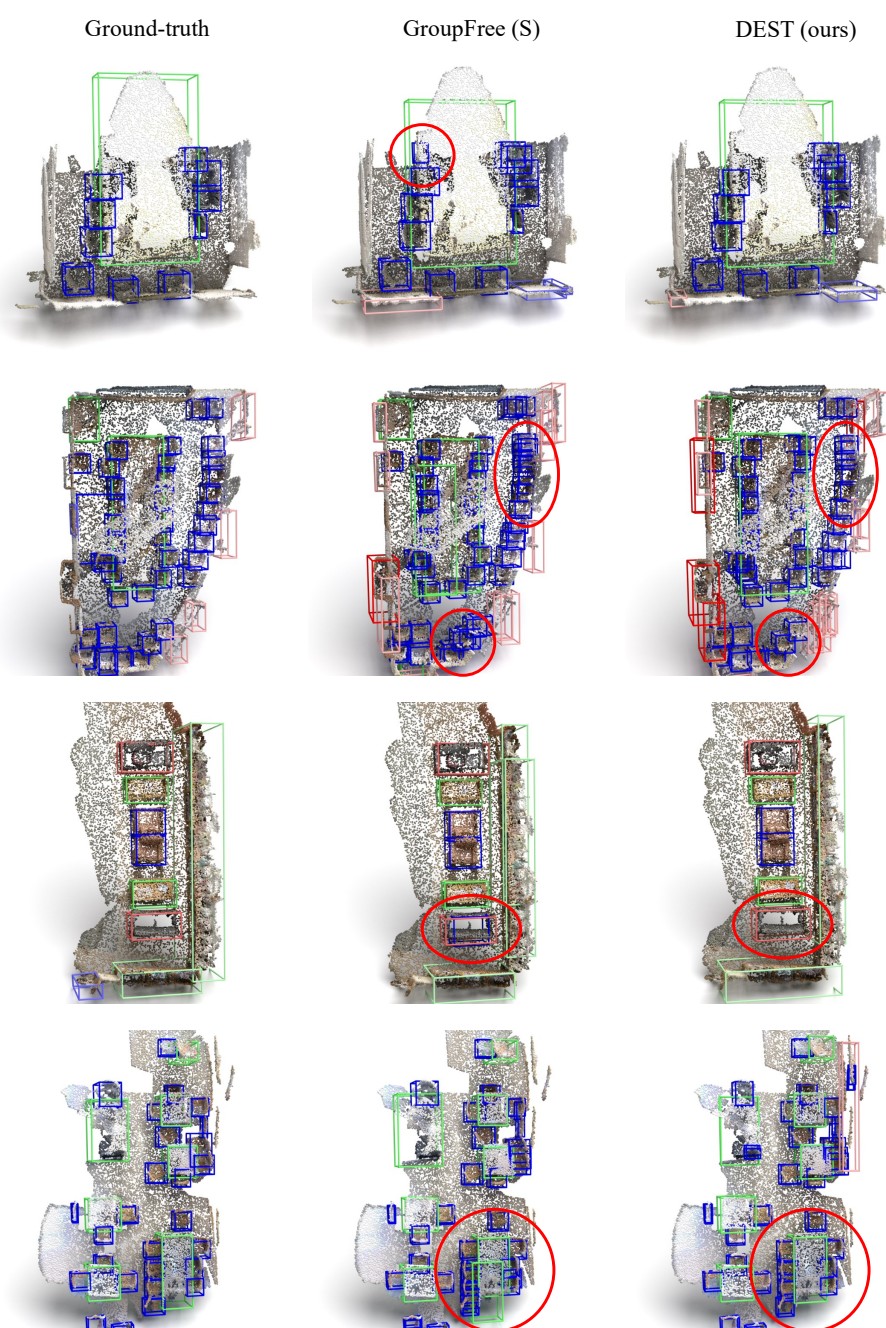

Figure 6: **Visual comparison with the GroupFree (S) baseline on ScanNet V2 dataset.** The ground truth is displayed in the first column, the baseline detection results in the second column, and our detection results in the third column.

## A.8 DISCUSSIONS ABOUT THE LIMITATIONS AND FUTURE RESEARCH

In this section, we discuss the limitations of this work and potential future research. We present a novel framework, DEST, for 3D object detection, which utilizes the proposed ISSM to achieve joint updates of scene points and query points during the decoding process. Although our method overcomes the performance bottleneck caused by fixed scene features in transformer decoders and significantly improves the performance of DETR-based methods, there are still instances of missed object

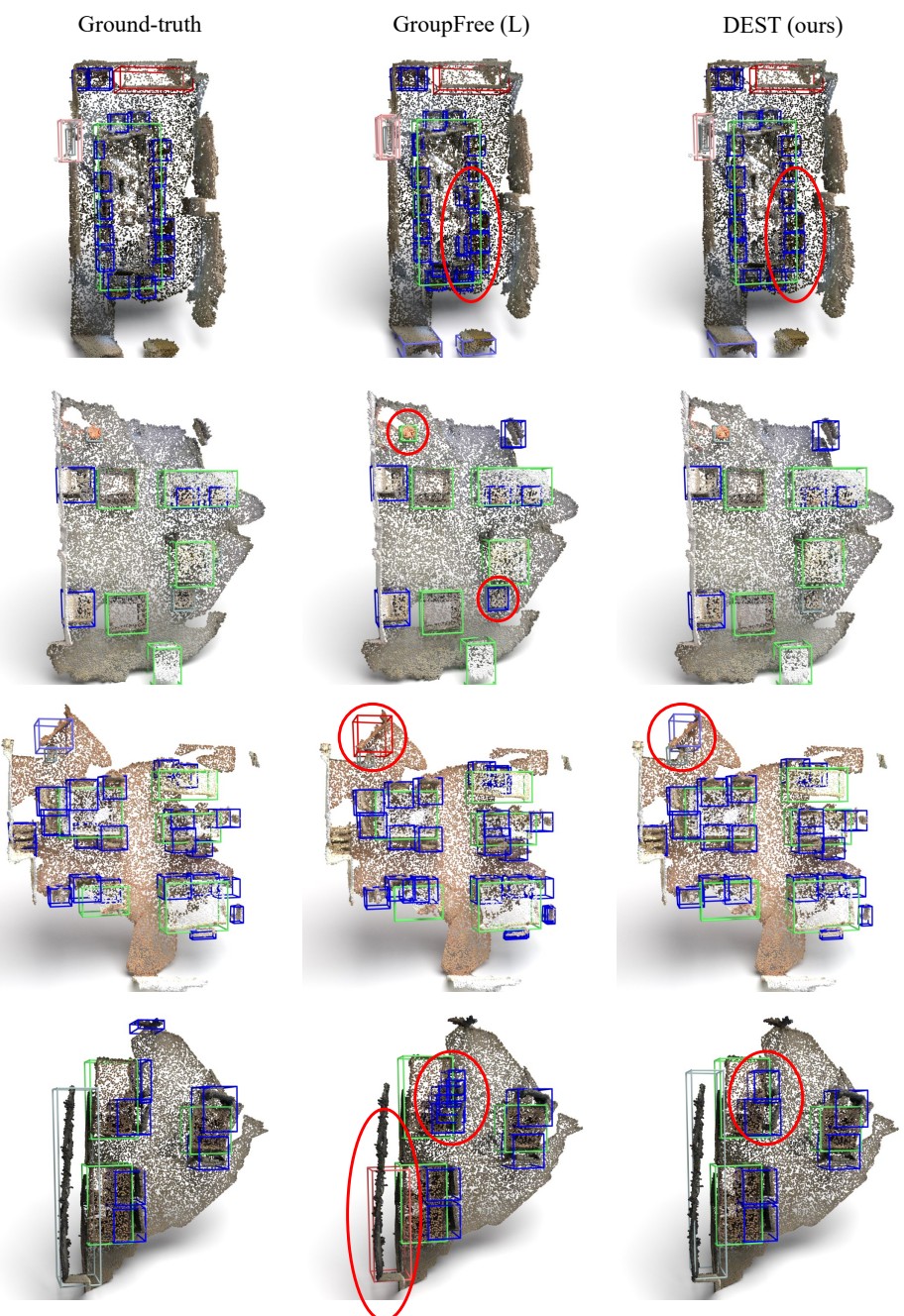

Figure 7: **Visual comparison with the GroupFree (L) baseline on ScanNet V2 dataset.** The ground truth is displayed in the first column, the baseline detection results in the second column, and our detection results in the third column.

detections. Through layer-by-layer analysis of the model, we find that the root cause of these missed detections is the absence of initial object candidate points. This study primarily focuses on the decoder design and does not delve into designing an initial state point sampling method. Therefore, designing a more effective state point sampling strategy to reduce the number of missed detections is a promising direction for future research. Beyond improving the proposed DEST framework, exploring its effectiveness in other tasks, such as 3D point cloud segmentation, tracking, and even 2D

Ground-truth        VDETR        DEST (ours)

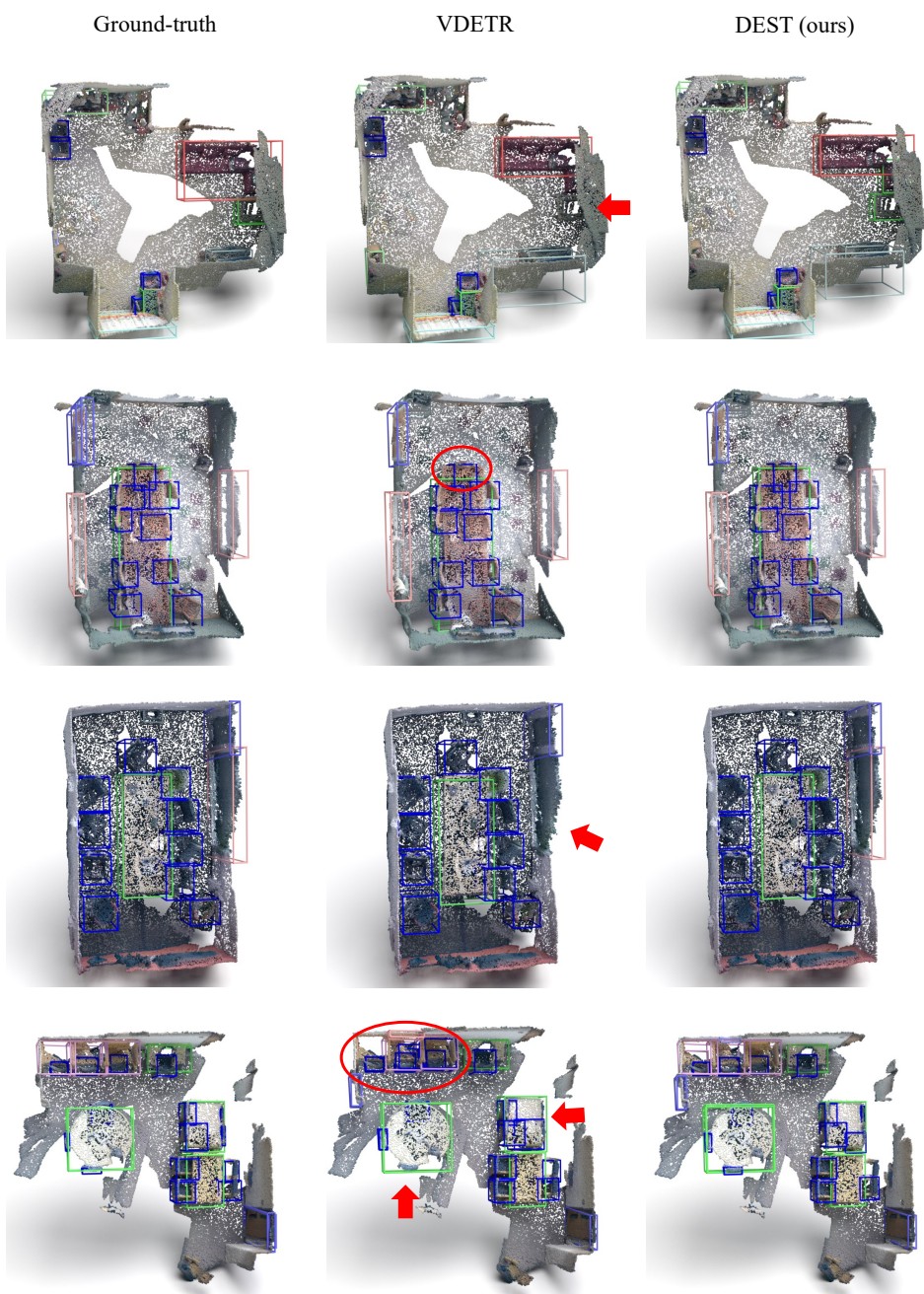

Figure 8: **Visual comparison with the VDETR baseline on ScanNet V2 dataset.** The ground truth is displayed in the first column, the baseline detection results in the second column, and our detection results in the third column.

vision tasks, is an interesting research direction. In the future, we will further explore the potential of the DEST framework in achieving unified point cloud perception tasks.

Ground-truth      GroupFree (S)      DEST (ours)

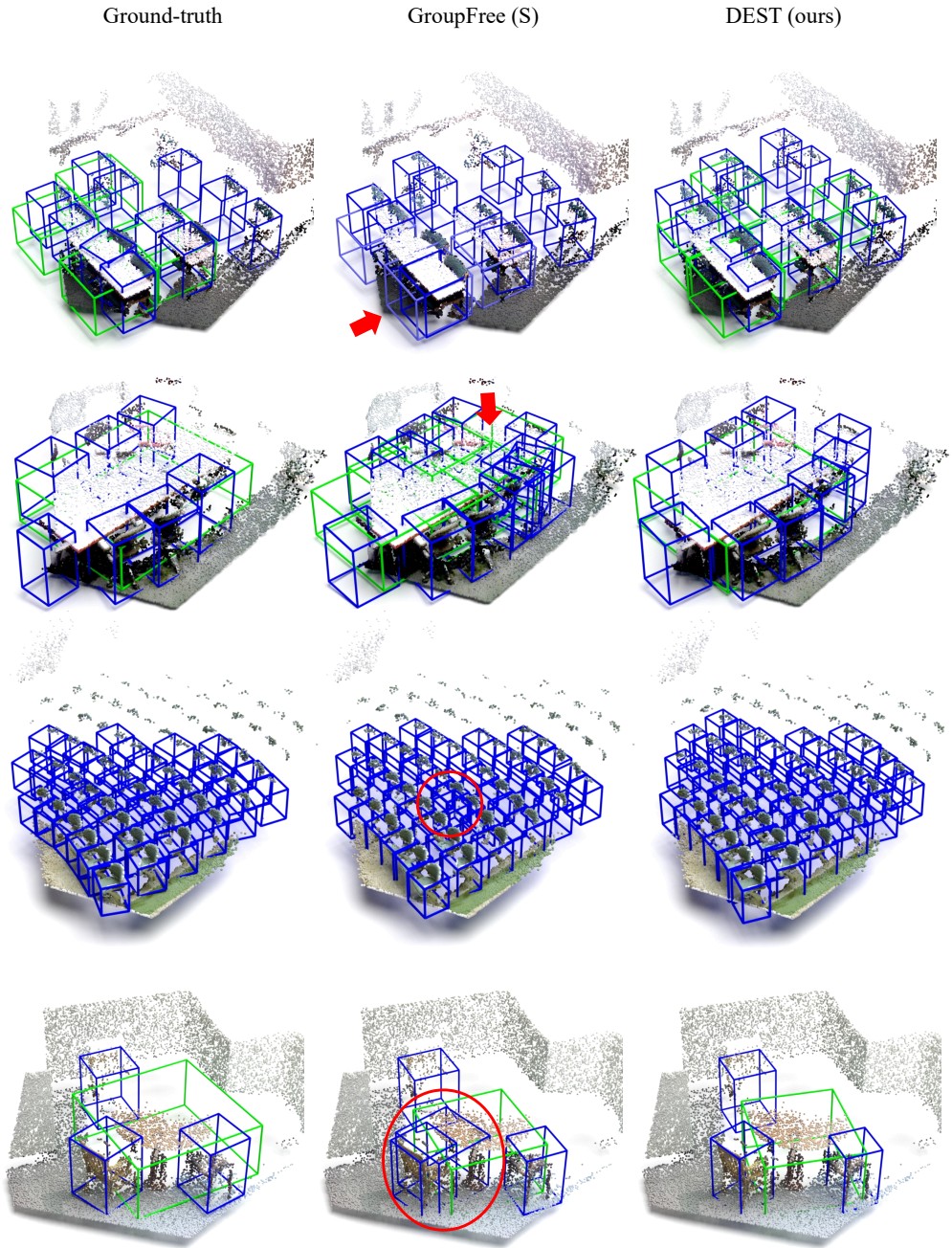

Figure 9: **Visual comparison with the GroupFree (S) baseline on SUN RGB-D dataset.** The ground truth is displayed in the first column, the baseline detection results in the second column, and our detection results in the third column.

