# OpenReview forum: "State Space Model Meets Transformer: A New Paradigm for 3D Object Detection"
_ICLR.cc/2025/Conference — ICLR 2025 Poster_

### Official Review · Reviewer_bWrW · 2024-10-29

**Soundness:** 4
**Presentation:** 3
**Contribution:** 3
**Rating:** 8
**Confidence:** 4

**Summary:**

Based on the motivation that sharing same scene representations in different decoder layers limits the overall performance of DETR-based 3D detectors, the paper proposes a mamba-based decoder architecture for indoor 3D detection. To fit the mamba design to 3D detection, the paper treats queries as states and scene representations as inputs. Experiments show that the proposed method improves existing baseline models.

**Strengths:**

1. The idea of updating scene features and query point features simultaneously is well-motivated.

2. With sufficient and extensive evaluations and ablations, the authors validate that DEST surpasses previous baseline methods with reasonable computational cost.

**Weaknesses:**

1. As mentioned by the authors in line 87 - 89, it is possible to "update the scene feature throughout the decoder layers with transformers". Compared to the input point cloud, processing the encoded points (1024/4096 for GroupFree3D/V-DETR) with quadratic time complexity is somewhat acceptable. It is also possible to implement the attention layers with modern flash-attention approaches [R1] for efficiency. The authors are encouraged to present numerical comparisons to support the choice for SSMs.

2. Missing citations. The Hilbert-based point cloud serialization strategy is closely related to PTv3[R2]. The authors are encouraged to state the relation to [R2].

[R1] Dao, Tri, et al. "Flashattention: Fast and memory-efficient exact attention with io-awareness." Advances in Neural Information Processing Systems 35 (2022): 16344-16359.

[R2] Wu, Xiaoyang, et al. "Point Transformer V3: Simpler Faster Stronger." Proceedings of the IEEE/CVF Conference on Computer Vision and Pattern Recognition. 2024.

**Questions:**

1. As shown in Fig. 4 (a), the difference between the proposed method mainly replaces the cross-attention layers in the standard transformer decoder architecture with the proposed ISSM blocks. Is it possible to also replace the inter-state attention module with bi-directional mamba?

---

> ### Author Response · Authors · 2024-11-21
>
> Thank you for your constructive review. We are glad that you find our idea well-motivated and experiments sufficient and extensive. Below are our responses to the weaknesses and questions:
>
>
>
> **W1. Numerical comparisons to support the choice for SSMs.**
>
> Thank you for the suggestion. We provide numerical comparisons with methods that update scene point features using attention layers. As shown in the following table, we use standard self-attention, FlashAttention [R1], and selective SSM to update the scene feature in the decoder layers. While attention-based methods improve the detection accuracy of baseline models, they still fall short compared to methods employing SSMs. This limitation arises because global attention is less effective at handling local relationships in point clouds and may introduce noise that disrupts scene point feature updates. In contrast, selective SSM uses 1D convolution to model local relationships and incorporates a hidden state selection mechanism to focus on extracting relevant features. Therefore, we adopt SSM to address the issue of fixed scene point features in the decoder. In our proposed DEST method, query points representing foreground objects serve as state points, providing richer and more task-specific foreground information to enhance scene point feature updates.
>
>  | Method | AP25 | AP50 |
>  |--------|------|------|
>  | GroupFree | 66.3 | 48.5 |
>  | w standard self-attention| 66.7 | 49.1 |
>  | w FlashAttention | 66.5 | 49.2 |
>  | w selective SSM | 67.0 | 50.1 |
>  | Ours | 67.9 | 52.7 |
>
>
>
> **W2. The relation of the serialization strategy in our method and PTv3 [R2].**
>
> We clarify the relationship between our serialization strategy and PTv3 [R2] and will include it in the revised manuscript.
>
> PTv3 employs z-order and Hilbert curves for point cloud serialization and introduces the shuffle order strategy to enhance the generalization of each attention layer. In our method, we generate six different serialized results for the point cloud by reordering the x, y, and z axes of the Hilbert curve, enabling more comprehensive observations. Unlike Serialized Attention in PTv3, random serialization disrupts sequential feature modeling in SSM and degrades performance. Thus, we apply distinct serialization orders across decoder layers without using the shuffle order strategy.
>
>
>
> **Q1. Replace the inter-state attention module with a bi-directional Mamba.**
>
> The inter-state attention module can be replaced with the bi-directional Mamba module, which models relationships among state points to capture inter-object dependencies within a scene. As shown in the following table, the bi-directional Mamba achieves performance comparable to the inter-state attention module.
>
>  | Method | AP25 | AP50 |
>  |--------|------|------|
>  | Ours | 67.9 | 52.7 |
>  | R Bi-Mamba | 67.8 | 52.5 |
>
>
>
> [R1] Dao, Tri, et al. "Flashattention: Fast and memory-efficient exact attention with io-awareness." Advances in Neural Information Processing Systems 35 (2022): 16344-16359.
>
> [R2] Wu, Xiaoyang, et al. "Point Transformer V3: Simpler Faster Stronger." Proceedings of the IEEE/CVF Conference on Computer Vision and Pattern Recognition. 2024.

---

> > ### Author Response · Authors · 2024-11-25
> >
> > We sincerely thank you for your time in reviewing our paper and your constructive comments. We have posted our point-to-point responses in the review system. Since the public discussion phase will end very soon, we appreciate if you could read our responses and let us know your feedback and further comments.

---

> > ### Comment · Reviewer_bWrW · 2024-11-26
> >
> > I thank the authors for their detailed response, which has addressed most of my concerns. I will maintain my positive score.
> >
> > Compared to the absolute performance, I am also interested in the computational efficiency, i.e. the training and inference time, memory cost of different design choices (bi-dir mamba vs attention).

---

> > > ### Author Response · Authors · 2024-11-26
> > >
> > > We sincerely appreciate your positive feedback and acceptance of our work. For the design choices (Bi-Mamba vs. Attention), we add the computational efficiency to the table in response to Q1. The training time and memory usage are measured with a batchsize of 8 per GPU, while the inference time and memory usage are measured with a batchsize of 1. All experiments are conducted on Tesla V100 GPUs. The model employing Bi-Mamba demonstrates notable advantages in computational efficiency, while the model utilizing Attention exhibits a relative advantage in detection accuracy.
> > >
> > > | Method | AP25 | AP50 | Training Time | Training Memory | Inference Time | Inference Memory |
> > > |--------|------|------|------------------------|---------------------|-------------------------|---------------------|
> > > | Ours | 67.9 | 52.7 | 885 ms/iter | 21.6 GB | 34 ms/sample | 3.14 GB |
> > > | R Bi-Mamba | 67.8 | 52.5 | 863 ms/iter | 20.1 GB | 33 ms/sample | 2.89 GB |

---

> > > > ### Comment · Reviewer_bWrW · 2024-11-27
> > > >
> > > > My concerns are addressed. I thank the authors again for their detailed response.

---

### Official Review · Reviewer_Y9XD · 2024-10-29

**Soundness:** 2
**Presentation:** 2
**Contribution:** 2
**Rating:** 6
**Confidence:** 4

**Summary:**

In this paper, the authors observe that the performance of DETR-based 3D object detection methods increases slowly in the last few Transformer decoder blocks. Based on the observation, this paper introduces state space models into 3D object detection and proposes an interactive state space model to update both 3D scene features and queries. Specifically, the authors design a state-dependent SSM parameterization to make each query interact with its respective scene features. Besides, the authors propose four key designs in an SSM-based decoder (*i.e.*, Hilbert-based serialization, inter-state attention, IBS module, and gated FFN). Experimental results show that the proposed method can be combined with multiple 3D detectors and further improve the detection accuracy on both Scannet V2 and SUN RGB-D datasets.

**Strengths:**

- The authors observe that the performance improves slowly in the last few blocks in the DETR-based 3D detector which is interesting.
- Experimental results show that the proposed methods can be combined with multiple DETR-based detectors and achieve SOTA performance.

**Weaknesses:**

- Questions about the proposed method
  - The first question is about computational complexity, in the proposed **Extension of $\Delta$**, the authors expand $\Delta$ $K$ times which will largely increase the computational complexity. The author should provide some results about the computational cost such as FLOPs.
  - Secondly, the design of **Spatial Correlation** Module, especially the relative offsets is similar to the 3D Vertex Relative Position Encoding used in VDETR.
  - Further, the ISSM-based decoder stacks lots of elements with few explanations. the authors should explain the importance of each component in the method section. Besides, more illustrations should be provided, for example, the structure of the inter-state attention module. Designs such as Hilbert serialization have been used in Point Transformer v3. The bidirectional scanning mechanism is also common in vision Mamba. The authors need to explain their rationale to avoid being mistaken for a simple stacking of modules.

- Questions about the experiments
  - In table 5, do the results in Param. represent the total parameters or the added parameters when using DEST?

**Questions:**

Lots of typos should be modified.
- Line 28, imporoves -> improves
- line 107, perpormance -> performance

Also, in line 84, 0.64 -> 6.4 according to figure 1.

The authors should carefully examine the article.

---

> ### Author Response · Authors · 2024-11-21
>
> Thank you for your detailed review. We hope our following responses can address your concerns.
>
> ---
>
> **Q1. The computational cost of Extension of $\Delta$.**
>
> The extension of $\Delta$ does not introduce significant computational overhead. The dimensions of the discrete SSM parameters $(\overline{A}，\overline{B})$ remain unchanged before and after extending $\Delta$. Therefore, the extension of $\Delta$ only affects the parameter generation without altering SSM processing. This is clarified in line 287 of the manuscript and Algorithm 1 in the Appendix.
>
> To further clarify the computational overhead of the proposed ISSM, we provide the FLOPs for both parameter generation and SSM processing. As shown in the following table, under the same model structure, the computational overhead of ISSM is only slightly higher than that of Selective SSM. Both Selective SSM and ISSM adopt the multi-head structure of Mamba2 with 32 heads, where the number of scene points is 1024, the number of state points is 256, and the feature dimension is 256.
>
> |    Method     | Parameter Generation (M) | SSM Processing (M) | Total (M) |
> | :-----------: | :----------------------: | :----------------: | :-------: |
> | Selective SSM |          285.21          |       224.26       |  509.47   |
> |  ISSM(Ours)   |          303.04          |       224.26       |  527.30   |
>
> ---
>
> **Q2. The motivation and design of the Spatial Correlation module.**
>
> The proposed ISSM needs to address a complex state update challenge: how to select appropriate scene points for updating state points. To tackle this, we develop the Spatial Correlation module, which leverages the relative positional relationships between scene and state points to select suitable scene points for state updates. Similar to VDETR, we employ vertex relative position encoding to determine the relationships between scene and state points. Unlike VDETR, which injects positional information into the attention map, we use it for generating SSM parameters $(B,C,\Delta)$. Additionally, we design a delay kernel for $\Delta$ to dynamically adjust the update magnitude of state points based on their distance from scene points.
>
> ---
>
> **Q3. Explain the importance of each component in the method section.**
>
> Thank you for your suggestion. We provide more illustrations about the inter-state attention module, the Hilbert serialization and the bidirectional scanning mechanism.
>
> **For the inter-state attention module**,  we focus on modeling feature interactions among state points. In 3D object detection, objects in a scene often exhibit strong correlations. For example, tables and chairs commonly appear together, while beds and toilets rarely coexist in the same room. To capture such relationships, we employ a standard self-attention mechanism for state points, enabling them to leverage scene semantics and enhance detection accuracy.
>
> **For the serialization strategy**, we generate six different serialized results for the point cloud by reordering the x, y, and z axes of the Hilbert curve, aiming for comprehensive observation of the point cloud. Compared to PTv3 [R1], which only swaps the x and y axes, our serialization results are more diverse. Additionally, unlike Serialized Attention in PTv3, the sequential feature modeling in SSM is more sensitive to changes in the serialization method. Thus, we apply distinct serialization orders across decoder layers without the Shuffle Order strategy used in PTv3.
>
> **For the bidirectional scanning mechanism**, we aim to model bidirectional point-to-point relationships in a single pass. We follow Vision Mamba [R2] and introduce the bidirectional scanning mechanism into our ISSM. Unlike the original Mamba, our ISSM uses scene points as system inputs and state points as system states. In the backward ISSM, we reverse the order of scene points while keeping the state points unchanged.
>
> ---
>
> **Q4. The meaning of "Param." in Table 5.**
>
> In Table 5, the results in "Param." represent the total parameters of our DEST-base model. We clarify this in the revised manuscript to avoid any ambiguity.
>
> ---
>
> **Q5. Typos.**
>
> We thoroughly revised the manuscript to correct typos and improve any unclear expressions.
>
> ---
>
> [R1] Wu, Xiaoyang, et al. "Point Transformer V3: Simpler Faster Stronger." Proceedings of the IEEE/CVF Conference on Computer Vision and Pattern Recognition. 2024.
>
> [R2] Zhu, Lianghui, et al. "Vision mamba: Efficient visual representation learning with bidirectional state space model." *arXiv preprint arXiv:2401.09417* (2024).

---

> > ### Comment · Reviewer_Y9XD · 2024-11-22
> > **Decision after rebuttal**
> >
> > The author solved my questions during the rebuttal phase. I will slightly improve my score.

---

> > > ### Author Response · Authors · 2024-11-22
> > >
> > > We sincerely thank this reviewer for the positive feedback and recognition of our work.

---

### Official Review · Reviewer_uYMo · 2024-11-03

**Soundness:** 3
**Presentation:** 3
**Contribution:** 4
**Rating:** 8
**Confidence:** 3

**Summary:**

The paper address an issue that the ﬁxed scene point features lead to suboptimal reﬁnement of query points in the later layers of transformer and then introduce a novel DEST-based method for 3D indoor object detection. The key design is ISSM, which models query points as system states and scene points as system inputs, allowing simultaneous updates with linear complexity. Experiments in ScanNetv2 and SUN RGB-D show its effectiveness.

**Strengths:**

1.	Meaningful topic：The paper starts from this problem that scene point features in the transformer decoder remain ﬁxed, leading to minimal contributions from later decoder layers, thereby limiting performance improvement for DETR-based methods.
2.	Convincing experiments: The paper tries its method on different baseline and datasets to verify its effectiveness. Ablation studies are also comprehensive.
3.	Creative design of method: The paper designs a novel state-dependent SSM parameterization method that enables system states to effectively serve as queries in 3D indoor detection tasks.

**Weaknesses:**

1.	For VDETR, improvement is not big enough.

**Questions:**

1.	Are there other methods to solve “fixed scene point” from previous works?
2.	Is is helpful to simultaneously update both scene point and query point features for most model, or just for DEST?
3.	State Space Models (SSM) has an efﬁcient context modeling ability with linear complexity through iterative interactions between system states and inputs. Is it possible to combine SSM and Transformers in 3D object detection?

---

> ### Author Response · Authors · 2024-11-21
>
> Thank you for your insightful review. We are glad that you find our topic meaningful, the experiments convincing, and the method design creative! Below are our responses to the questions:
>
> ---
>
> **Q1. Other methods to solve "fixed scene point" from previous works.**
>
> To the best of our knowledge, this study is the first to identify and address the performance limitations in 3D detection caused by fixed scene point features. However, some techniques from previous works could be adapted to address this issue. A straightforward approach involves introducing a module in the decoder to update scene point features, such as using standard self-attention, FlashAttention, or selective SSM.
>
> The following table shows that incorporating a feature-updating module in the decoder improves detection performance. However, the performance gains of these methods remain limited compared to our proposed method. We attribute this to the unique design of our method, where query points representing foreground objects serve as state points, providing richer and more task-specific foreground information to enhance scene point feature updates effectively.
>
> | Method                    | AP25 | AP50 |
> | ------------------------- | ---- | ---- |
> | GroupFree                 | 66.3 | 48.5 |
> | w standard self-attention | 66.7 | 49.1 |
> | w FlashAttention          | 66.5 | 49.2 |
> | w selective SSM           | 67.0 | 50.1 |
> | Ours                      | 67.9 | 52.7 |
>
> ---
>
> **Q2. Simultaneously updating scene and query point features is helpful for most DETR-based models.**
>
> Simultaneously updating is beneficial for most DETR-based models for the following two reasons:
>
> 1. DETR-based methods commonly use the multi-layer transformer decoder to refine query points iteratively. The issue of fixed scene point features limits the performance improvement from later decoder layers.
> 2. Updating scene point features in decoder provides more adaptive and context-rich information to the query points, enhancing their refinement process and ultimately improving detection accuracy.
>
> The experimental results in Q1 also validate that simultaneously updating scene and query point features benefits the DETR-based models, even when using a simple global attention mechanism.
>
> ---
>
> **Q3. Combine SSM and Transformers in 3D object detection?**
>
> Combining SSM and Transformer in 3D object detection is indeed feasible. As shown in the table of Q1, we incorporate selective SSM into the transformer decoder to update scene point features. This simple combination method improves the detection accuracy, and it holds potential for further exploration in the future.

---

> > ### Author Response · Authors · 2024-11-25
> >
> > We sincerely thank you for your time in reviewing our paper and your constructive comments. We have posted our point-to-point responses in the review system. Since the public discussion phase will end very soon, we appreciate if you could read our responses and let us know your feedback and further comments.

---

> > > ### Comment · Reviewer_uYMo · 2024-11-26
> > > **Response to author rebuttal**
> > >
> > > The responses of authors address most of my concerns, so I will maintain my positive rating.

---

> > > > ### Author Response · Authors · 2024-11-26
> > > >
> > > > Thank you for your positive feedback! We're delighted to hear that our responses address your concerns and appreciate your recognition of our work.

---

### Official Review · Reviewer_Ho6d · 2024-11-11

**Soundness:** 2
**Presentation:** 3
**Contribution:** 3
**Rating:** 6
**Confidence:** 2

**Summary:**

The paper presents a novel approach to 3D object detection, particularly for indoor environments, by integrating State Space Models (SSMs) with Transformer architectures. The key contributions of the paper are:

1. **Introduction of DEST (State Space Model-based 3D Object Detection)**: The paper proposes a new paradigm called DEST, which addresses the limitations of fixed scene point features during the query refinement process in existing methods. This is the first method to model queries as system states within an SSM framework.

2. **Design of an Interactive State Space Model (ISSM)**: The authors develop an ISSM that effectively functions as queries in complex 3D indoor detection tasks. The ISSM-based decoder is tailored to the characteristics of 3D point clouds, fully harnessing the potential of the ISSM for 3D object detection.

3. **Enhanced Performance**: Extensive experimental results demonstrate that the proposed SSM-based 3D object detection method consistently enhances the performance of baseline detectors on two challenging indoor datasets, ScanNet V2 and SUN RGB-D. Comprehensive ablation studies validate the effectiveness of each designed component.

4. **ISSM-based Decoder**: The paper introduces an ISSM-based decoder that replaces the transformer decoder in DETR-based methods, addressing the performance limitations caused by fixed scene point features. This decoder simultaneously updates both scene point and query point features.

5. **Comparison with State-of-the-Art Methods**: The DEST-based detector significantly outperforms previous 3D object detection methods on the ScanNet V2 and SUN RGB-D datasets, demonstrating substantial performance improvements.

Overall, the paper introduces a new state-space model-based approach to 3D object detection that overcomes the limitations of existing methods, particularly in handling complex indoor environments, and achieves state-of-the-art performance.

**Strengths:**

1. Writing: The writing is clear and effectively conveys the content of the paper, making it easy to understand.

2. Innovative SSM Integration: The paper creatively combines State Space Models with transformers, offering a unique approach to 3D object detection that reduces complexity while enhancing performance.

3. Performance Improvement: The model achieves substantial performance gains on benchmarks, specifically outperforming existing models on AP metrics for key datasets (ScanNet V2 and SUN RGB-D).

4. Detailed Experimentation: Extensive experiments, including ablation studies, provide strong evidence for the model’s effectiveness and validate each component’s contribution.

**Weaknesses:**

1. According to Table 5, the proposed method results in a noticeable increase in latency.
2. What is the motivation for introducing SSM in this paper? Why is SSM suitable for 3D detection tasks? I believe this is a critical point that requires detailed explanation from the authors.

**Questions:**

Please refer to section Weaknesses.

---

> ### Author Response · Authors · 2024-11-21
>
> Thank you for your detailed review. We are glad that you find our paper well-written and our method innovative! Below are our responses to the weaknesses and questions:
>
> ---
>
> **Q1. The motivation for introducing SSM in this paper.**
>
> We introduce SSM with the following two motivations:
>
> 1. To integrate modules within the decoder for updating scene point features, thereby overcoming the performance bottleneck caused by fixed scene point representations in DETR-based methods.
>
> 2. To avoid the substantial computational and memory overhead associated with self-attention mechanisms when applied to updating dense scene point features.
>
> ---
>
> **Q2. Why is SSM suitable for 3D detection tasks?**
>
> Applying SSM to 3D detection tasks offers the following three advantages:
>
> 1. **Simultaneous Update of Scene and Query Point Features**: By treating query points as system states and scene points as system inputs, SSM enables simultaneous feature updates of query points and scene points.
> 2. **Effective Feature Modeling**: As detailed in Appendix A.1, state points in SSM effectively extract information from scene points to achieve accurate detection. Additionally, scene points leverage the rich foreground information encoded in state points to update their features.
> 3. **Linear Complexity**: Compared to transformers, SSM significantly reduces computational and memory overhead, making it well-suited for handling dense scene points.
>
> ---
>
> **W1. The proposed method results in a noticeable increase in latency.**
>
> The need for multiple updates to scene point features in the decoder inevitably introduces some latency. Although we employ an SSM with linear complexity to minimize computational overhead, certain delays are unavoidable. Considering the substantial performance gains, this trade-off is acceptable.

---

> > ### Author Response · Authors · 2024-11-25
> >
> > We sincerely thank you for your time in reviewing our paper and your constructive comments. We have posted our point-to-point responses in the review system. Since the public discussion phase will end very soon, we appreciate if you could read our responses and let us know your feedback and further comments.

---

> > ### Comment · Reviewer_Ho6d · 2024-11-27
> > **Response to author rebuttal**
> >
> > The responses of authors address most of my concerns. I will maintain my rating.

---

### Meta-Review · Area_Chair_ZEAe · 2024-12-22

**Metareview:**

This paper integrates State Space Models (SSMs) with Transformer architectures for the 3D object detection task, and shows promising results on ScanNetV2 and SUN RGB-D datasets. The manuscript was reviewed by four experts in the field. The recommendations are (2 x "6: marginally above the acceptance threshold", 2 x "8: accept, good paper"). Based on the reviewers' feedback, the decision is to recommend the acceptance of the paper. The reviewers did raise some valuable concerns (especially additional and important experimental evaluations and ablation studied, needed comparisons with previous literature, detailed clarification on statements, and together with further polishment of the manuscript) that should be addressed in the final camera-ready version of the paper. The authors are encouraged to make the necessary changes to the best of their ability.

**Additional Comments On Reviewer Discussion:**

Reviewers mainly hold the concern regarding extra additional and important experimental evaluations and ablation studies (Reviewer bWrW), needed comparisons with previous literature (Reviewers uYMo, bWrW), detailed clarification on statements (Reviewers Ho6d, uYMo), and together with further polishment of the manuscript (Reviewer Y9XD). The authors address these concerns with detailed and extra experiments and commit to polishing the revised version further.

---

### Decision · Program_Chairs · 2025-01-22

Accept (Poster)